# Angiotensin-(3–4) normalizes the elevated arterial blood pressure and abnormal Na⁺/energy handling associated with chronic undernutrition by counteracting the effects mediated by type 1 angiotensin II receptors

Amaury Pereira-Acácio[1,2,3,4], João P. M. Veloso-Santos[3,4], Luiz F. Nossar[3], Gloria Costa-Sarmento[3,4], Humberto Muzi-Filho[3,4], Adalberto Vieyra[1,3,4,5]*

1 Graduate Program of Translational Biomedicine/BIOTRANS, University of Grande Rio, Duque de Caxias, Brazil, 2 Leopoldo de Meis Institute of Medical Biochemistry, Federal University of Rio de Janeiro, Rio de Janeiro, Brazil, 3 Carlos Chagas Filho Institute of Biophysics, Federal University of Rio de Janeiro, Rio de Janeiro, Brazil, 4 National Center of Structural Biology and Bioimaging/CENABIO, Federal University of Rio de Janeiro, Rio de Janeiro, Brazil, 5 National Institute of Science and Technology for Regenerative Medicine/REGENERA, Rio de Janeiro, Brazil

* avieyra@biof.ufrj.br

**Data Availability Statement:** All relevant data are within the paper.

## Abstract

We investigated the mechanisms by which chronic administration of a multideficient diet after weaning alters bodily Na⁺ handling, and culminates in high systolic blood pressure (SBP) at a juvenile age. From 28 to 92 days of age, weaned male Wistar rats were given a diet with low content and poor-quality protein, and low lipid, without vitamin supplementation, which mimics the diets consumed in impoverished regions worldwide. We measured food, energy and Na⁺ ingestion, together with urinary Na⁺ excretion, Na⁺ density (Na⁺ intake/energy intake), plasma Na⁺ concentration, SBP, and renal proximal tubule Na⁺-transporting ATPases. Undernourished rats aged 92 days had only one-third of the control body mass, lower plasma albumin, higher SBP, higher energy intake, and higher positive Na⁺ balance accompanied by decreased plasma Na⁺ concentration. Losartan or Ang-(3–4) normalized SBP, and the combination of the 2 substances induced an accentuated negative Na⁺ balance as a result of strong inhibition of Na⁺ ingestion. Na⁺ density in undernourished rats was higher than in control, irrespective of the treatment, and they had downregulated (Na⁺+K⁺)ATPase and upregulated Na⁺-ATPase in proximal tubule cells, which returned to control levels after Losartan or Ang-(3–4). We conclude that Na⁺ density, not only Na⁺ ingestion, plays a central role in the pathophysiology of elevated SBP in chronically undernourished rats. The observations that Losartan and Ang-(3–4) normalized SBP together with negative Na⁺ balance give support to the proposal that Ang II⇒AT₁R and Ang II⇒AT₂R axes have opposite roles within the renin-angiotensin-aldosterone system of undernourished juvenile rats.

**Funding:** This work is supported by Brazilian Research Council/CNPq (https://www.gov.br/cnpq/pt-br, grants 470266/2014-7 and 440544/2018-1 to AV), Rio de Janeiro State Foundation/FAPERJ (https://www.faperj.br/, grants E- 26/210.890/2019 and E-26/201.909/2020 to AV), and the Brazilian Federal Agency for Support and Evaluation of Graduate Education/CAPES (https://www.gov.br/capes/pt-br, grants 88887.124150/2014-00 to AV, 88887.320213/2019-00 to HM-F, 88887.374390/2019-00 to AP-A and 88887.634142/2021-00 to AP-A). The funders had no role in study design, data collection and analysis, decision to publish, or preparation of the manuscript.

**Competing interests:** The authors have declared that no competing interests exist.

## Introduction

Undernutrition is characterized by insufficient ingestion of the necessary substances in both quantity and quality for growth and health, or by an inability to absorb or convert nutrients [1]. Nutritional imbalances can be associated with an upregulated renin-angiotensin-aldosterone system (RAAS) leading to cardiorenal pathologies, including hypertension [2], in which altered mechanisms of $Na^+$ handling in the kidney and the heart are important, together with modifications of type 1 and type 2 angiotensin II receptor ($AT_1R$ and $AT_2R$) signaling [3]. Although there is growing information regarding the association between undernutrition, RAAS, and altered cardiovascular regulation [4], the interactions between the two main axes of RAAS (Ang II⇒$AT_1R$ and Ang II⇒$AT_2R$) in chronic undernutrition remain to be studied. Knowledge concerning the influence of these axes on specific parameters in undernutrition, and the chronic undernutrition-induced modifications in renal $Na^+$-transporting ATPases and renal and bodily $Na^+$ balance is particularly lacking.

We induced chronic undernutrition with a multideficient diet that was administered to male Wistar rats after weaning. This diet (Regional Basic Diet/RBD) mimics those used in vast impoverished regions of undeveloped countries, and also in pockets of poverty in big cities worldwide [5]. It has served for many years as a model for the study of many more pathologies in different organs and tissues [6]. Its protein content is low and of poor quality; lipids, vitamins, and minerals are also low. Teodósio and coworkers [5] formulated this diet according to the dietary habits of populations from vast sugarcane regions in Northeast Brazil. The change caused by RBD in the body weight of rats resembles that encountered in humans [5].

The rats were exposed to this diet after weaning until 92 days of age aiming to cover a period that, in humans, corresponds to the nursing period, the prepubescent period, the adolescent period, and the beginning of adulthood [7]. As pointed out 7 years ago [4], this model reflects the situation encountered in underdeveloped countries, where undernutrition starts after an appropriate nutrition period by breastfeeding. The first pharmacological intervention—to shed light on the role of RAAS—was designed to inhibit the Ang II⇒$AT_1R$ axis for the entire period of dietary deficiency. This approach aimed to investigate the influence of $AT_1R$ signaling blockade in the evolution of arterial pressure and, at the end of the period, i.e. at the beginning of adulthood, to have a picture of the influence of RAAS on $Na^+$ handling and fluid balance in undernourished rats. Blockade of the Ang II⇒$AT_1R$ axis was achieved by the administration of Losartan, an antagonist of $AT_1R$ [8]. The second pharmacological intervention determined whether acute activation of the Ang II⇒$AT_2R$ axis at the young adult age modifies arterial pressure and influences $Na^+$ handling.

Ang II⇒$AT_2R$ axis was stimulated by administration of Ang-(3–4) (Val-Tyr), the shortest Angiotensins-derived peptide, which antagonizes most of the Ang II actions [9–11] by acting as an allosteric enhancer of Ang II binding to $AT_2R$ [12]. The choice was based on early observations regarding the antihypertensive effects of Ang-(3–4) in humans [9] and in spontaneously hypertensive rats (SHR); in SHR, Ang-(3–4) stimulates urinary excretion of $Na^+$ [10]. We demonstrated that Ang-(3–4) acts in pro-hypertensive tissular microenvironments [10, 13], i.e. in tissues in which the activity of local RAAS is high [14].

## Materials and methods

### Animal care

All experimental procedures were approved by the Committee for Ethics in Animal Experimentation of the Federal University of Rio de Janeiro (protocols 007/16 and 012/19 prepared and registered before the study), and were performed in accordance with the Committee's

guidelines, which follow the Uniform Requirements for Manuscripts Submitted to Biomedical Journals. The animal study is reported in accordance with ARRIVE guidelines [15]. The rats were maintained in the vivarium at 22 ± 2˚C under a 12 h dark:12 h light cycle, continuous air renewal throughout the study, and veterinary control.

## Experimental groups

Female Wistar rats were kept and mated in the Vivarium for Study of Neglected Diseases and Undernutrition (Carlos Chagas Filho Institute of Biophysics, Federal University of Rio de Janeiro). The animals resulted from 4 successive breedings and they were allocated to different sets of experiments. A random-number table was used to allocate the rats and only 2 observers (AP-A and JPMV-S) were aware of the group allocation. These 2 observers were in charge of the order and registration of treatments and determinations. The rats were individualized with numbers on the tail using indelible ink, and clear labels in the cages avoided any possibility of mistaken identity. The total number of animals was 153. Not all animals were used in each class of experiments because, in several cases, the experiments were planned and performed based on the results obtained using rats from one previous breeding. This is the reason why the exact "n" values were different in each experiment, as indicated in the figure legends. The size of the samples was calculated according to [16] for continuous variables.

Male offspring were weaned at 28 days of age and the total number of groups at the end of the study was 8. Initially (at the age of 28 days) the animals were randomly divided into 2 groups: the first group received the CTRL diet, and the other group received the multideficient diet RBD (Table 1) until the end of the study. On the same day, each of the 2 groups was randomly subdivided into 2 further subgroups, thereby originating groups that received Losartan (Los: 30 mg/kg body mass diluted in the drinking water; Biosintética, Jurubatuba, Brazil), daily from weaning to 91 days of age, thus starting the additional CTRL+Los and RBD+Los groups. This dose was chosen because it reduces blood pressure and prevents renal injury in spontaneously hypertensive rats when daily administered [18].

At the age of 91 days, a subgroup of each 4 groups received vehicle (water) or one single oral dose of Ang-(3–4) (80 mg/kg body mass; EZBiolab, Carmel, IN, USA) by gavage. Thus,

**Table 1. Composition of diets.**

|  | CTRL[1] | RBD[2] |
|---|---|---|
| Protein % (w/w) | 23 | 8 |
| Carbohydrate % (w/w) | 41 | 78 |
| Lipids % (w/w) | 2.5 | 1.7 |
| Na % (w/w) | 0.3[3] | 0.2[3] |
| Fe % (w/w) | 0.018 | 0.007 |
| Ca % (w/w) | 1.8 | 0.04 |
| K % (w/w) | 0.9 | 0.3 |
| Energy supply kcal/100 g dry weight | 278 | 356 |
| Vitamin supplement | Yes | No |

CTRL, Control diet; RBD, Regional Basic Diet.

[1] As indicated by the manufacturer (Neovia Nutrição e Saúde Animal, São José do Vale do Rio Preto, Brazil).

[2] According to the Laboratory of Experimental and Analysis of Food (LEEAL), Nutrition Department, Federal University of Pernambuco.

[3] According to Muzi-Filho *et al.* [17].

the 4 new groups were now: CTRL+Ang-(3–4) (CTRL rats treated with Ang-(3–4)); CTRL +Los+Ang-(3–4) (CTRL+Los rats treated with Ang-(3–4)); RBD+Ang-(3–4) (RBD rats treated with Ang-(3–4)) and RBD+Los+Ang-(3–4) (RBD+Los rats treated with Ang-(3–4)). Each rat was the experimental unit in the study. In the case of $Na^+$-transporting ATPases, the experimental units were different membrane preparations obtained from a pool of kidneys from 3 (CTRL groups) or 5 rats (RBD groups).

Before and after 24 h of Ang-(3–4) administration, the rats were placed in metabolic cages to measure food intake and for recording urinary volume. Before and after the metabolic cage period, the blood pressure of the rats from the 8 groups was measured. The rats were decapitated for plasma collection and kidney dissection to obtain plasma membrane preparations from proximal tubules for the *in vitro* experiments (see below). During the whole period of the experimental protocol, food and filtered water were available *ad libitum*. Note: the experimental period of the 4 groups of animals treated with Ang-(3–4) lasted one extra day because they derived from the other 4 groups.

Appropriate titles in the figure legends clearly explain the aim of the comparisons among groups within each panel of the figures.

## Blood pressure measurements

Systolic blood pressure was measured by a non-invasive method [19] in conscious rats at day 91 by using a tail-cuff plethysmograph (model V2.01; Insight, Ribeirão Preto, Brazil). An additional record on day 92 was carried out in the groups that received Ang-(3–4). Digital signals were recorded and processed by using the appropriate software (Pressure Gauge 1.0, Insight). On the day before the procedure, the rats were acclimated in a heated chamber (30–32ºC) for 10 to 15 min, and the recordings were only taken from the rats without sudden movements. Five determinations were made for each animal and the average of the 5 values was used.

## Preparation of plasma membrane-enriched fraction from kidney proximal tubule cells

Membrane preparations were obtained by homogenization and differential centrifugations from the outermost region of the renal cortex (*cortex corticis*) [20], where the cell population corresponds to >95% of proximal tubules [21]. Preparation of the plasma membrane-enriched fraction from kidney proximal tubule cells and control for the residual contamination with membranes from intracellular organelles (succinate dehydrogenase activity for mitochondria and glucose-6-phosphatase for endoplasmic reticulum) was as previously described [20], with slight modifications. Briefly, thin transverse slices of the *cortex corticis* (0.5 mm) were separated with a Stadie-Riggs microtome (Thomas Scientific, Swedesboro, NJ, USA), immersed in an isotonic solution containing 10 mM Hepes-Tris (pH 7.4), 250 mM sucrose, 2 mM EDTA, 1 mM PMSF and 0.15 mg/mL trypsin inhibitor type II-S (T1021; Sigma-Aldrich, St. Louis, MO, USA), and dissected using a small ocular scissor. The fragments were homogenized at 4ºC in the same isotonic solution (1 g tissue/4 mL solution) using a Potter Elvejhem homogenizer fitted with a Teflon pestle (5 cycles of 1 min at 1700 rpm). The resulting homogenate went through 3 successive differential centrifugations: (i) $10000 \times g$ for 15 min at 4ºC (JA-20 rotor, Beckman Avanti J-E centrifuge; Beckman Coulter, Brea, CA, USA), (ii) $15000 \times g$ for 20 min at 4ºC (JA-20 rotor, Beckman Avanti J-E centrifuge), and (iii) $35000 \times g$ for 44 min at 4ºC (70 Ti rotor, Beckman Optimal L-90K ultracentrifuge). The pellets were resuspended in 250 mM sucrose to ~15 mg/mL protein, quantified by the Folin phenol method, using bovine serum albumin as standard [22], aliquoted into tubes, and stored at -80ºC. The enrichment factor for

ouabain-sensitive (Na$^+$+K$^+$)ATPase, used as a marker for basolateral membranes, was 5–6 with respect to the initial homogenate. The contamination with mitochondria and endoplasmic reticulum usually varied around 3% and 5%, respectively. The samples were used to measure the activity of the 2 Na$^+$-transporting ATPases, the ouabain-sensitive (Na$^+$+K$^+$)ATPase and the ouabain-resistant Na$^+$-ATPase, as described below.

### Albumin and Na$^+$ determinations

Plasma albumin was measured with a commercial kit (catalog number: K040, Quibasa-Bioclin, Belo Horizonte, Brazil). Na$^+$ concentrations in urine and plasma samples were determined by flame photometry (Analyzer, São Paulo, Brazil) using a standard solution containing 140 mequiv Na$^+$/L (Analyzer).

### Determination of the activities of Na$^+$-transporting ATPases from kidney proximal tubules

The ouabain-sensitive (Na$^+$+K$^+$)ATPase and the ouabain-resistant, furosemide-sensitive Na$^+$-ATPase activities were determined by quantifying the inorganic phosphate (P$_i$) released during ATP hydrolysis [23]. (Na$^+$+K$^+$)ATPase activity was measured in plasma membranes from proximal tubules (0.025 mg/mL, final concentration), which were preincubated in the absence or presence of 2 mM ouabain (Sigma-Aldrich) for 10 min at 37ºC in a medium (0.5 mL) containing 50 mM Bis-Tris-Propane (pH 7.4), 0.2 mM EDTA, 5 mM MgCl$_2$ and 120 mM NaCl. The reaction was started by simultaneous addition of 24 mM KCl and 5 mM ATP (final concentrations), and stopped 10 min later by adding 0.5 mL of 0.1 M HCl-activated charcoal. The suspension was centrifuged (13300 × g for 10 min) and part of the supernatant was diluted with the same volume of a solution containing 0.2 N H$_2$SO$_4$, 10 mM ammonium molybdate, and 0.3 M FeSO$_4$; absorbance was recorded at 660 nm 20 min later.

Ouabain-resistant Na$^+$-ATPase activity was measured in the same membrane preparations (0.05 mg/mL, final concentration), which were preincubated in the presence of 2 mM ouabain, in the presence or absence of 2 mM furosemide (Sigma-Aldrich) for 10 min at 37ºC in 20 mM Hepes-Tris (pH 7.0), 10 mM MgCl$_2$ and 120 mM NaCl. Assays started by adding 5 mM ATP in a final volume of 0.5 mL and stopped with 0.5 mL 0.1 M HCl-activated charcoal before being processed as described for (Na$^+$+K$^+$)ATPase. The activities were calculated by: (i) the differences between the values obtained in the absence and presence of 2 mM ouabain for (Na$^+$+K$^+$)ATPase; (ii) the difference between the values obtained in the absence and presence of 2 mM furosemide for Na$^+$-ATPase [14, 15]. Determinations were carried out in triplicate.

### Statistical analysis

Statistical analyses were carried out using GraphPad Prism 6 software (version 6.01, GraphPad Software, Inc., San Diego, CA, USA). Results are expressed as mean ± standard error of the mean (SEM). Differences were assessed using one-way ANOVA followed by Bonferroni's test for selected pairs (see figure legends). Significant differences were set at $p < 0.05$; ns when $p \geq 0.05$. P values are given within the panels. The 2 rate constants of growth $k$ (CTRL and RBD rats) were compared using the Student´s $t$-test.

## Results

### Chronic blockade of type 1 Ang II receptors (AT$_1$R) differentially modified evolution of body mass in CTRL and RBD rats

The body mass (BM) evolution of normonourished CTRL (empty circles in Fig 1) and undernourished RBD rats (empty squares in Fig 1) was followed using the equation:

$$BM_t = BM_{28} + BM_{max} \times (1 - e^{-kt}) \tag{1}$$

where $BM_t$ corresponds to BM at different times $t$ (days), $BM_{28}$ corresponds to BM at weaning (64 ± 2 g, considering all rats used), $BM_{max}$ is the theoretical value of body mass gain attained at time → ∞ from the departure value $BM_{28}$, $k$ is the rate constant of growth and $e$ has the usual meaning. The average $k$ values were different (0.0346 ± 0.0020 *vs* 0.0203 ± 0.0010 days$^{-1}$ for RBD and CTRL rats, respectively; $t$ = 6.95, p < 0.0001, unpaired Student's $t$-test), indicating: (i) that the undernourished rats grew daily at a more significant fraction of their reduced $BM_{max}$ (48 g in RBD *vs* 327 g in CTRL rats); (ii) that the time required to attain 50% of their reduced $BM_{max}$ ($t_{1/2}$) was lower (20 and 34 days, respectively). Chronic administration of RBD culminated in a 70% decrease in body mass (BM; 97 ± 3 g) at 92 days compared with CTRL rats (306 ± 5 g) given commercial chow.

Blocking AT$_1$R by chronic administration of Losartan from weaning differentially modified the profile of growth in CTRL and RBD groups: though the body mass gain was lower in both groups, the corresponding profiles were different (compare the evolution of filled symbols in Fig 1). In the CTRL animals, 2 different phases of growth—and, therefore, 2 different time-dependencies—were clearly seen when Losartan was given to CTRL rats. AT$_1$R blockade changed the kinetics of growth over the first period of 28 days of diet administration, following

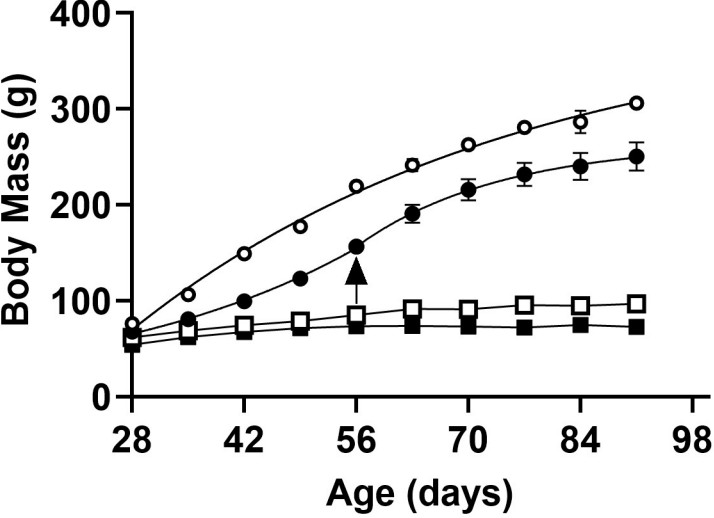

**Fig 1. Body mass (BM) development.** Rats received the control diet (CTRL) (circles) or the multideficient diet (RBD) (squares) for 64 days from weaning at 28 days until 92 days of age: effect of Losartan (Los) administration for the same period indicated on the *abscissae*. Dietary composition is described in Table 1. Empty symbols: untreated rats; filled symbols: Losartan-treated rats. Data are mean ± SEM; n = 5 (CTRL), n = 5 (CTRL+Los), n = 8 (RBD), n = 6 (RBD +Los). In several cases, error bars are smaller than the symbol size. Eqs 1 and 2 were adjusted to the experimental points, the kinetic parameters of growth being described in the text and summarized in Table 2. Differences among the four groups were assessed weekly using one-way ANOVA followed by Bonferroni's test. Statistical difference was set at p < 0.05. The differences appeared from the 1$^{st}$ week (CTRL *vs*. RBD) and from the 2$^{nd}$ week (CTRL *vs*. CTRL+Los). No differences were found between the RBD and the RBD+Los groups. The arrow indicates the transition between the 2 different phases of growth in Losartan-treated CTRL rats.

the exponential function:

$$BM_t = BM_{28} + e^{kt} \qquad (2)$$

where the factors have the same meaning as above, and $k = 0.1545 \pm 0.0027$ days$^{-1}$, an apparently faster rate constant. However, in real nutritional terms, this kinetic behavior meant an accentuated decrease in growth at early ages (compare the empty and filled circles in Fig 1). Interestingly, the profile rapidly changed from the initial period and onwards, recovering that described by Eq 1, with a faster $k$ ($0.0574 \pm 0.0054$ days$^{-1}$) ($t_{1/2} = 12$ days) but always at reduced BM compared to untreated rats (compare the empty and filled circles in Fig 1), attaining a theoretical additional mass gain $BM_{max} = 110$ g. In Losartan-treated RBD rats, the growth curve followed the same single function seen in untreated rats (filled squares in Fig 1), attaining a lower body mass gain ($BM_{max} = 19$ g) with a faster $k$ ($0.1079 \pm 0.0058$ days$^{-1}$) and a very low $t_{1/2}$ (6 days). Table 2 gives a simpler comparison of the kinetic parameters of growth corresponding to the 4 groups.

The undernutrition status of rats given RBD is also reflected in the plasma albumin ([albumin]$_{pls}$), with a 40% decrease with respect to CTRL, without an effect of Losartan in both groups (Fig 2A). Acute administration of Ang-(3–4) alone provoked an accentuated hypoalbuminemia in CTRL and RBD rats (Fig 2B); however, when the 2 drugs were given together to the CTRL group, the trend was attenuated and reversed in the case of RBD rats to the CTRL +Los+Ang-(3–4) values (Fig 2C) ($p \geq 0.05$).

## Food and energy intake in CTRL and RBD rats that received Losartan and Ang-(3–4)

The next figures show the data regarding food and energy intake. Food intake in 24 h per 100 g BM was 30% higher in RBD rats at day 92, 24 h after they had been acclimated in individual cages (Fig 3A). Losartan decreased food intake in RBD, but not in the CTRL group (Fig 3A), and a fall in food ingestion was provoked by Ang-(3–4) in both groups, being quite remarkable in undernourished rats (Fig 3B). Combined administration of the drugs (Fig 3C) decreased feeding of CTRL, but not of RBD rats. Both treatments recovered the CTRL food intake in the undernourished rats. When the effects of the combined treatments were compared with the respective untreated CTRL and RBD groups, a significant and accentuated anorexigenic effect was seen in both groups (Fig 3D).

**Table 2. Kinetic parameters of BM evolution from 28 (weaning) to 92 days of age.**

|  | Additional body mass gain $BM_{max}$ (g) | Rate constant of growing $k$ (days$^{-1}$) | Time to 50% of $BM_{max}$ $t_{1/2}$ (days) |
|---|---|---|---|
| CTRL | 327 | 0.0203 | 34 |
| CTRL +Los |  |  |  |
| 1$^{st}$ phase | * | 0.1545 | ** |
| 2$^{nd}$ phase | 110 | 0.0574 | 12 |
| RBD | 48 | 0.0346 | 20 |
| RBD+Los | 19 | 0.1079 | 6 |

* The theoretical $BM_{max}$ of this phase does not have biological/nutritional meaning because $BM_{max} \to \infty$ when $t \to \infty$.

** The same reasoning applies for the calculated $t_{1/2} = 4$ days in the 1$^{st}$ phase.

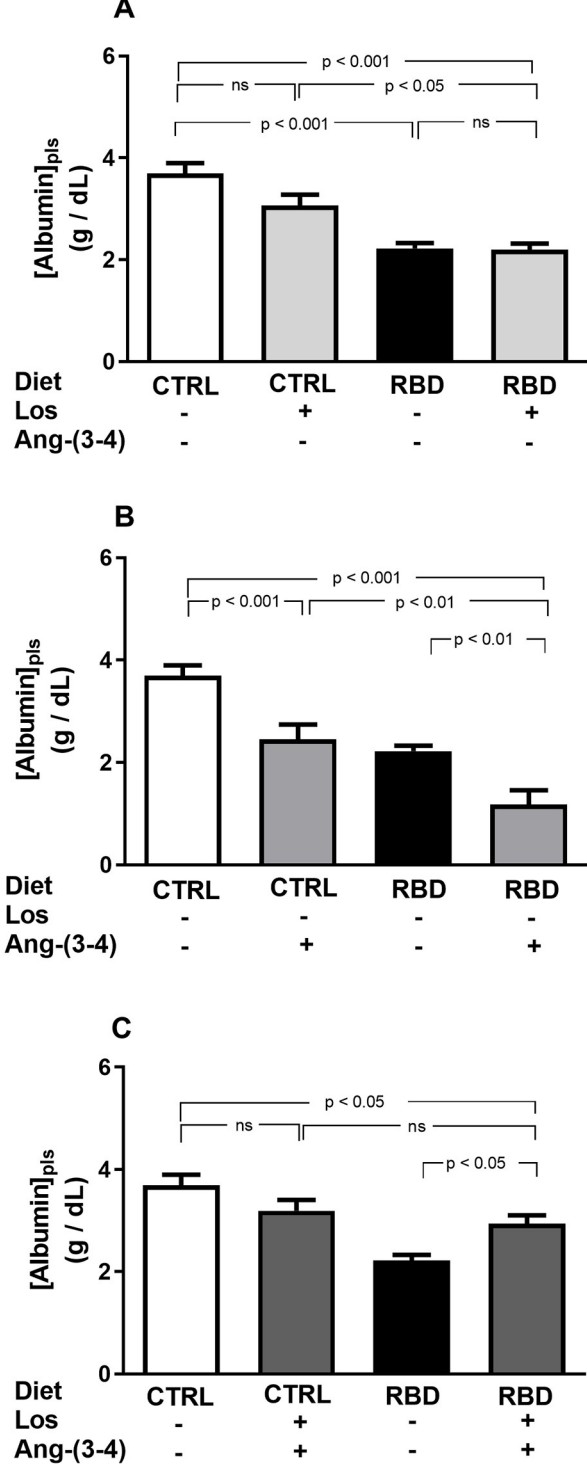

**Fig 2. Responses of plasma albumin concentration to Losartan and Ang-(3–4) in normonourished and undernourished rats. (A)** Responses to Losartan. Comparison of plasma albumin concentration in CTRL and RBD rats without chronic administration of Losartan, and effects of Losartan administration (30 mg/kg body mass per day, from 28 to 92 days of life) to CTRL and RBD rats, as indicated on the *abscissa*. **(B)** Responses to Ang-(3–4). Effects of oral administration of a single dose of Ang-(3–4) (80 mg/kg body mass), at 91$^{st}$ day of life, to CTRL and RBD rats, as indicated on the *abscissa*. **(C)** Responses to Losartan+Ang-(3–4). Effects of combined administration of Losartan and Ang-(3–4) to CTRL and RBD rats, at the doses above indicated, as shown on the *abscissa*. Samples were collected at 91 days, except for Ang-(3–4)-treated rats, which were collected at 92 days. Bars indicate mean ± SEM; n = 24 (CTRL),

n = 28 (RBD), n = 10 (CTRL+Los), n = 21 (RBD+Los), n = 16 (CTRL+Ang-(3–4)), n = 13 (RBD+Ang-(3–4)), n = 17 (CTRL+Los+Ang-(3–4)), n = 19 (RBD+Los+Ang-(3–4)). Differences were assessed using one-way ANOVA followed by Bonferroni's test for selected pairs. P values are given within the panels.

The energy intake of the RBD rats was 100% higher than in CTRL and it was decreased by Losartan (30%) (Fig 4A) and Ang-(3–4) (35%) in the RBD group (Fig 4B) with a small, but significant, influence of Ang-(3–4) in CTRL (Fig 4B). In contrast to that found with dietary intake, the drugs did not help in recovering the CTRL energy intake (compare Fig 3A and 3B with Fig 4A and 4B, respectively) certainly because the decrease in food did not suffice to compensate for the higher caloric content of the deficient diet. A combination of treatments

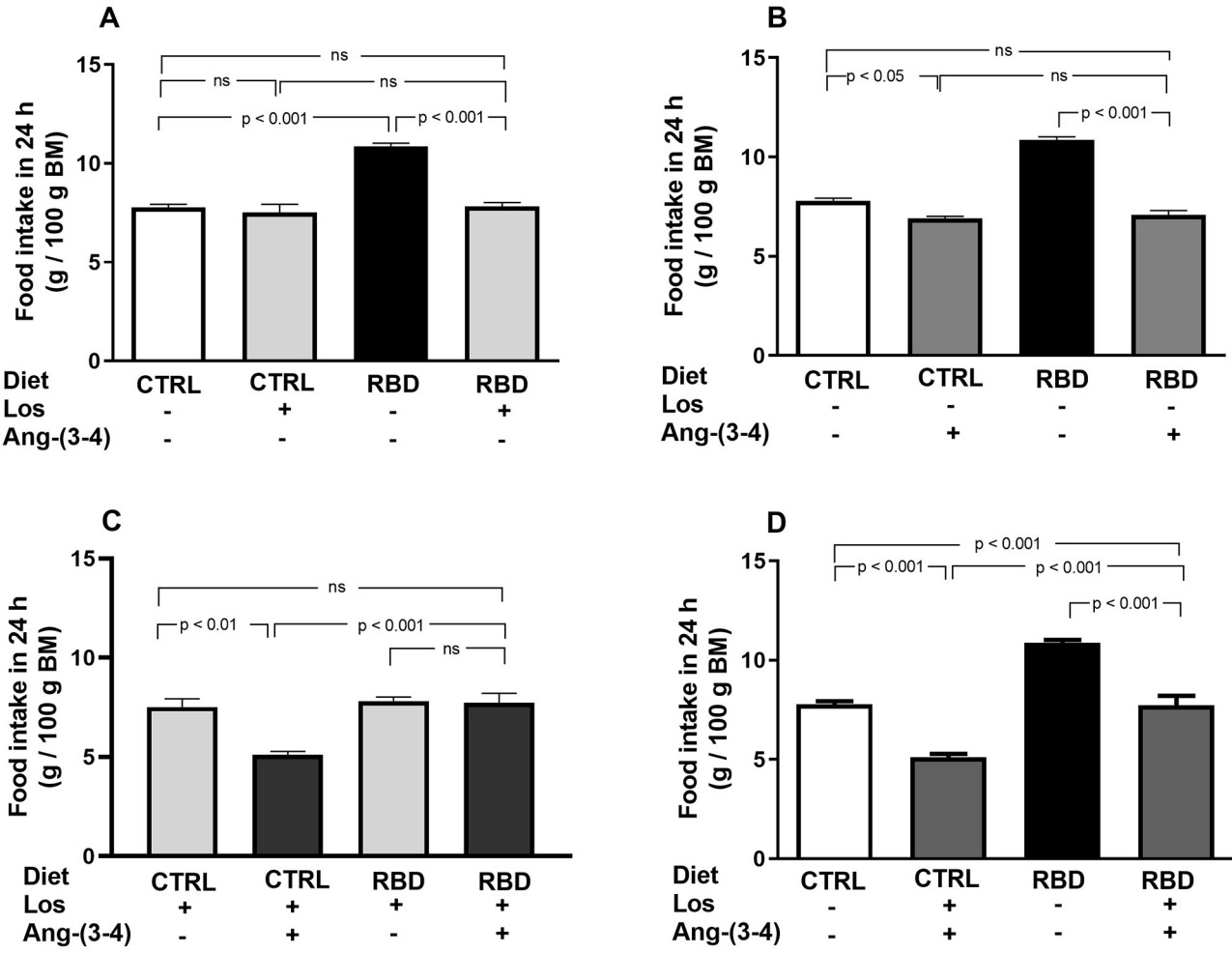

Fig 3. Food intake. (A) Responses to Losartan. Comparison of food intake by CTRL and RBD rats without chronic administration of Losartan, and effects of Losartan administration, as indicated on the *abscissa*. (B) Responses to Ang-(3–4). Effects of oral administration of a single dose of Ang-(3–4) on food intake by CTRL and RBD rats, as indicated on the *abscissa*. (C) Responses to Ang-(3–4) in rats previously treated with Losartan. Effects of oral administration of a single dose of Ang-(3–4) on food intake by CTRL and RBD rats previously treated with Losartan, as indicated on the *abscissa*. (D) Effects of combined Losartan+Ang-(3–4) administration. Comparison of food intake between untreated CTRL and RBD rats *vs*. CTRL and RBD rats that were chronically given Losartan and a single dose of Ang-(3–4), as indicated on the *abscissa*. Food intake was measured between 90 and 91 days, except for Ang-(3–4)-treated rats, for which determinations were carried out between 91 and 92 days. Bars are mean ± SEM (data expressed per 100 g BM); n = 5 (CTRL), n = 8 (RBD), n = 5 (CTRL+Los), n = 6 (RBD+Los), n = 5 (CTRL+Ang-(3–4)), n = 8 (RBD+Ang-(3–4)), n = 5 (CTRL+Los+Ang-(3–4)), n = 6 (RBD+Los+Ang-(3–4)). Differences were assessed using one-way ANOVA followed by Bonferroni's test for selected pairs. P values are indicated within the panels.

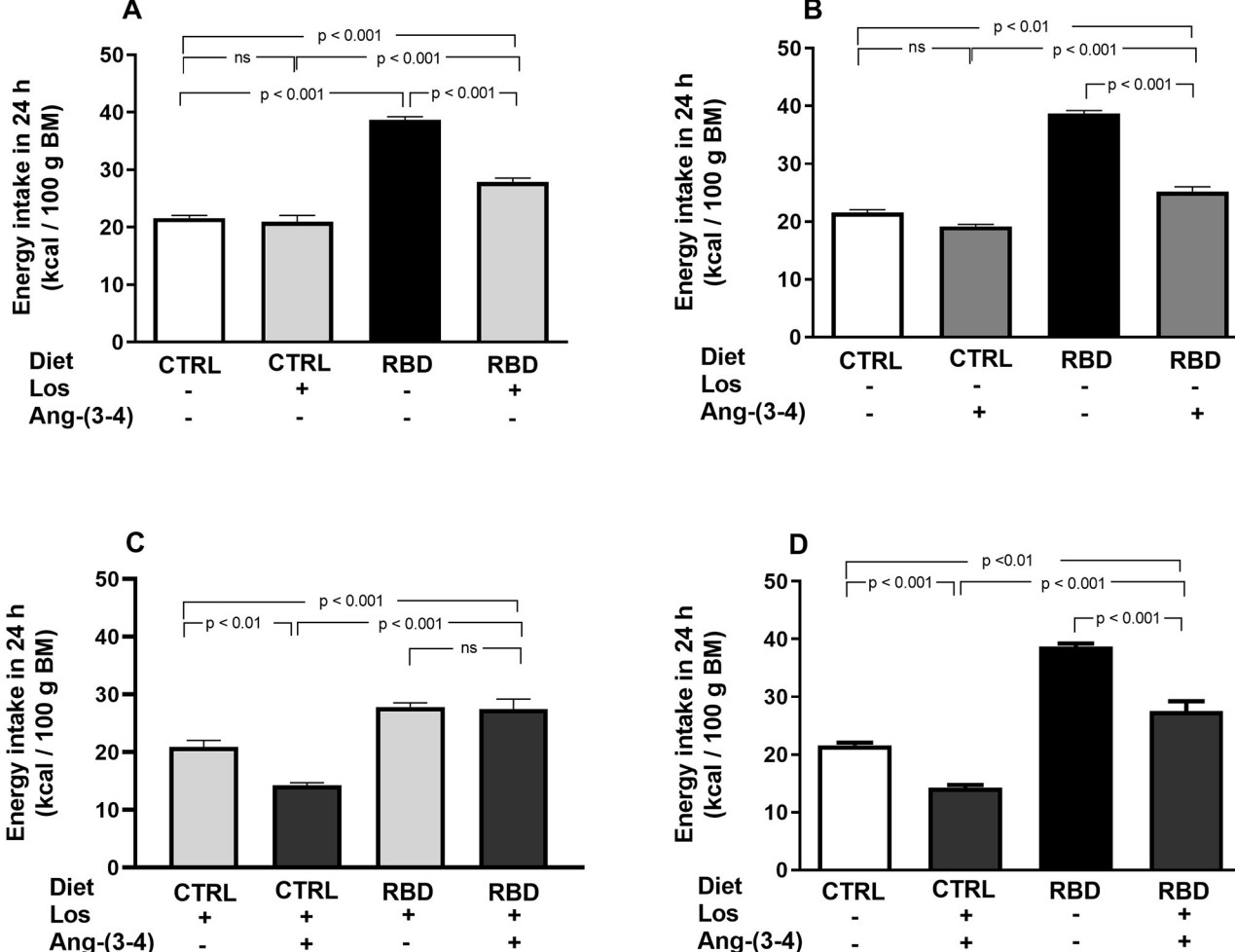

**Fig 4. Energy intake.** (**A**) Responses to Losartan. Comparison of energy intake by CTRL and RBD rats without chronic administration of Losartan, and effects of Losartan administration, as indicated on the *abscissa*. (**B**) Responses to Ang-(3–4). Effects of oral administration of a single dose of Ang-(3–4) on energy intake by CTRL and RBD rats, as indicated on the *abscissa*. (**C**) Responses to Ang-(3–4) in rats previously treated with Losartan. Effects of oral administration of a single dose of Ang-(3–4) on energy intake by CTRL and RBD rats previously treated with Losartan, as indicated on the *abscissa*. (**D**). Responses of combined Losartan+Ang-(3–4) administration. Comparison of energy intake between untreated CTRL and RBD rats *vs*. CTRL and RBD that were chronically given Losartan and a single dose of Ang-(3–4), as indicated on the *abscissa*. Energy intake was calculated from food intake and the diet composition described in Table 1. Bars are mean ± SEM (data expressed per 100 g BM); n = 5 (CTRL), n = 8 (RBD), n = 5 (CTRL+Los), n = 6 (RBD+Los), n = 5 (CTRL+Ang-(3–4)), n = 8 (RBD+Ang-(3–4)), n = 5 (CTRL+Los+Ang-(3–4)), n = 6 (RBD+Los+Ang-(3–4)). Differences were assessed using one-way ANOVA followed by Bonferroni's test for selected pairs. P values are indicated within the panels.

accentuated the decrease in energy intake by CTRL rats without further influence in the RBD group (Fig 4C); they also resulted in the pronounced diminution of the caloric ingestion compared with the untreated CTRL and RBD groups (Fig 4D).

## Na⁺ intake, Na⁺ density, arterial hypertension, and Na⁺ balance in CTRL and RBD rats: Losartan- and Ang-(3–4)-induced modifications

To assess the influence of chronic undernutrition and the response to drug treatments on bodily Na⁺ handling, we measured Na⁺ intake, urinary Na⁺ concentration ([Na⁺]$_{ur}$), daily urinary volume, and Na⁺ excretion in 24 h (U$_{Na}$V) at 92 days of age. RBD rats ingested a small higher, but only a tendency, amount of Na⁺ per 100 g BM compared with CTRL animals

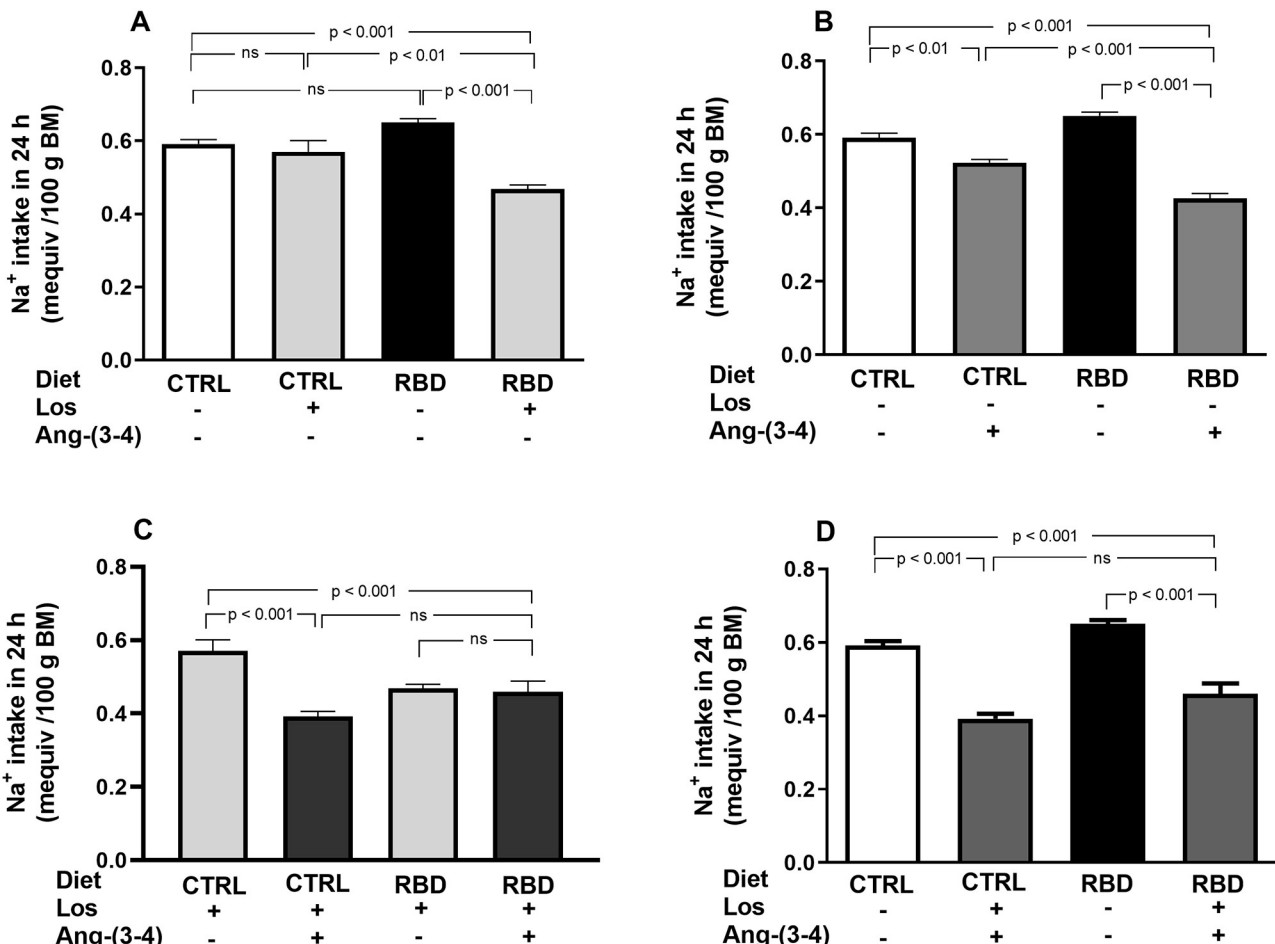

**Fig 5. Na+ intake.** Na+ intake in 24 h was calculated from the dietary Na+ content and the food intake, measured between 90 and 91 days, except for Ang-(3–4)-treated rats which were measured at 91 and 92 days. **(A)** Responses to Losartan. Comparison of Na+ intake by CTRL and RBD rats without chronic administration of Losartan, and effects of Losartan administration, as indicated on the *abscissa*. **(B)** Responses to Ang-(3–4). Effects of oral administration of a single dose of Ang-(3–4) on Na+ intake by CTRL and RBD rats, as indicated on the *abscissa*. **(C)** Responses to Ang-(3–4) in rats previously treated with Losartan. Effects of oral administration of a single dose of Ang-(3–4) on Na+ intake by CTRL and RBD rats previously treated with Losartan, as indicated on the *abscissa*. **(D)** Effects of combined Losartan+Ang-(3–4) administration. Comparison of Na+ intake between untreated CTRL and RBD rats *vs*. CTRL and RBD rats that were chronically given Losartan and a single dose of Ang-(3–4), as indicated on the *abscissa*. Bars are mean ± SEM (data expressed per 100 g BM); n = 5 (CTRL), n = 8 (RBD), n = 5 (CTRL+Los), n = 6 (RBD+Los), n = 5 (CTRL+Ang-(3–4)), n = 8 (RBD +Ang-(3–4)), n = 5 (CTRL+Los+Ang-(3–4)), n = 6 (RBD+Los+Ang-(3–4)). Differences were assessed using one-way ANOVA followed by Bonferroni's test for selected pairs. P values are indicated within the panels.

(Fig 5A) as a result of the increased intake of food (Fig 3A). Losartan and Ang-(3–4) had different influences depending on the nutritional condition, and also when given alone or in combination. When the drugs were given alone: (*i*) no effect of Losartan was seen in Na+ ingestion by the CTRL group, whereas in contrast there was a strong inhibition of ~30% in the RBD group (Fig 5A); (*ii*) Ang-(3–4) alone inhibited Na+ ingestion in both groups, which was more accentuated in RBD (35%) than in CTRL rats (10%) (Fig 5B). The administration of a single dose of Ang-(3–4) after chronic treatment with Losartan (Fig 5C) potentiated the inhibition in CTRL, but showed that the peptide did not modify the effect of Losartan in RBD rats. A comparison of untreated groups with respect to those receiving the combined treatment led to a similar 30% inhibition of Na+ intake by the drugs (Fig 5D).

Na$^+$ density is emerging as a concept defined as the ratio between Na$^+$ and energy in a diet [24] and, therefore, between Na$^+$ and energy intake. This correlation is presented in Fig 6 for the 8 experimental groups at day 92 of age, after the administration of vehicle or Ang-(3–4) to CTRL and RBD rats previously treated either with or without Losartan. Two straight lines were seen when the individual data of energy intake were plotted as a function of the corresponding Na$^+$ intake: one of the steepest slope corresponds to the 4 groups of RBD rats, and one of the lesser slope includes the 4 groups of CTRL rats. Moreover, 3 clusters can be identified: the frame ① shows that all untreated RBD rats were clustered above the cut-off levels of 35 kcal in 24 h per 100 g BM (horizontal dashed line) and 0.60 mequiv Na$^+$ ingested in 24 h per 100 g BM (vertical dashed line); the frame ② includes all the CTRL rats that received Ang-(3–4) and Losartan+Ang-(3–4) within a cut-off delimited by 20 kcal in 24 h/100 g BM (horizontal dashed line) and 0.55 mequiv Na$^+$ in 24 h/100 g BM (vertical dashed line); the frame ③ corresponds to the remaining panel, including (i) the RBD rats treated with Losartan, Ang-(3–4) or its combination, and (ii) the CTRL rats treated or untreated with Losartan.

An important characteristic of the RBD phenotype is arterial hypertension (Fig 7). The following results demonstrate that alterations in energy and Na$^+$ intake encountered in RBD rats are associated with important cardiovascular alterations. The systolic blood pressure (SBP) reached 150 mmHg in contrast with the 120 mmHg value for CTRL rats at 92 days of age, and the pressoric values were reversed by Losartan (totally) or partially by (Ang-(3–4) (Fig 7A and 7B). Interestingly, combined treatment resulted in similar SBP when the groups RBD+Ang-(3–4) and RBD+Los+Ang-(3–4) were compared (133.2 ± 0.6 and 133.9 ± 1.3 mmHg; Fig 7B and 7C, respectively). In other words, the combined treatment Los+Ang-(3–4) resulted in a reduced hypotensive effect than in the case of Losartan administration alone (119.1 ± 0.2 mmHg).

To calculate the U$_{Na}$V (and then the Na$^+$ balance besides Na$^+$ intake) at day 92 of age, we measured urinary volume over 24 h and [Na$^+$]$_{ur}$ (Fig 8). The urinary flux of RBD was higher than in CTRL rats and increased further in the animals that received Losartan, whereas no effect of the AT$_1$R antagonist was encountered in the CTRL group (Fig 8A). Ang-(3–4) alone decreased urinary flux of RBD rats to the levels of CTRL, which have not been modified by the peptide, whereas it decreased the urine volume in CTRL and RBD rats previously treated with Losartan (Fig 8B and 8C). The comparison of combined treatments with their respective untreated CTRL and RBD showed no significant differences (Fig 8D).

[Na$^+$]$_{ur}$ was lower in RBD rats compared to the CTRL, without further effect of Losartan in both groups (Fig 9A). Administration of Ang-(3–4) alone did not modify [Na$^+$]$_{ur}$ in both groups (Fig 9B), as well as the combined treatment with respect to Losartan alone (Fig 9C). A comparison of the combined treatments with the respective untreated groups also showed no differences (Fig 9D).

U$_{Na}$V values are given in Fig 10, demonstrating similar values in CTRL and RBD rats (Fig 10A), despite an increased Na$^+$ intake seen in Fig 5A for the undernourished animals; the effect of Losartan was different depending on the nutritional status: increased U$_{Na}$V only in RBD rats. Ang-(3–4) administration and the combined treatment had no effect on both groups of rats treated or not with Losartan (Fig 10B–10D).

With these U$_{Na}$V values and those of Na$^+$ intake above (Fig 5), we calculated the Na$^+$ balance (Fig 11), which showed that: (*i*) the positive Na$^+$ balance increased by ~100% in RBD rats compared with the CTRL group (Fig 11A, 1$^{st}$ *vs.* 2$^{nd}$ bars); (*ii*) the positive balance strongly decreased in Losartan-treated CTRL rats (Fig 11A, 5$^{th}$ bar; 11B, 2$^{nd}$ bar); (*iii*) RBD rats were insensitive to the drug (Fig 11C, 1$^{st}$ *vs.* 2$^{nd}$ bars), and consequently the difference between the CTRL and RBD groups increased to 400% (Fig 11A, 3$^{rd}$ *vs.* 4$^{th}$ bars); (*iv*) after Ang-(3–4) administration, the positive Na$^+$ balance approached zero in CTRL and was strongly depressed

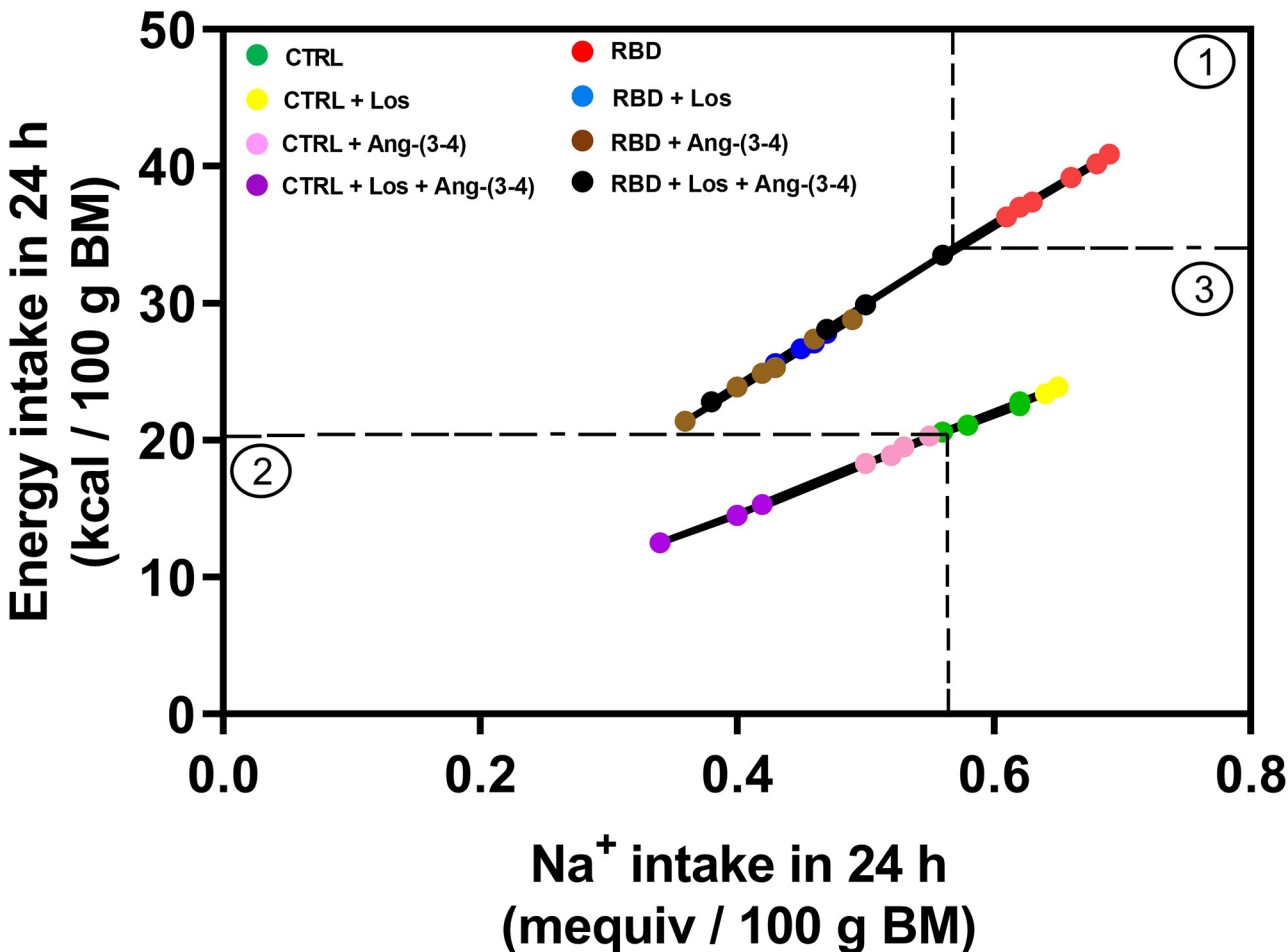

**Fig 6. Na⁺ density.** Correlation between Na⁺ intake and energy intake. Data points were calculated from the corresponding values given in Figs 5 and 6; n = 5 (CTRL), n = 8 (RBD), n = 5 (CTRL+Los), n = 6 (RBD+Los), n = 5 (CTRL+Ang-(3–4)), n = 8 (RBD+Ang-(3–4)), n = 5 (CTRL+Los+Ang-(3–4)), n = 6 (RBD+Los+Ang-(3–4)). The dashed lines delimit the 3 different clusters of Na⁺ densities (circles) described in the text. Upper line: RBD (red), RBD+Los (blue), RBD+Ang-(3–4) (brown), and RBD+Los+Ang-(3–4) (black). Bottom line: CTRL (green), CTRL+Los (yellow), CTRL+Ang-(3–4) (pink), and CTRL+Los+Ang-(3–4) (lilac). The function Energy intake = slope × Na⁺ intake was adjusted to the points by the least squares method. Upper line: slope = 59.2 kcal/mequiv of Na⁺; r = 0.9996. Bottom line: slope = 36.9 kcal/mequiv of Na⁺; r = 0.9994.

in RBD rats (Fig 11A, 5th *vs.* 6th bars; 11B, 3rd bar; 11C, 3rd bar); (*v*) with the combined treatment, Na⁺ balance became negative in the CTRL (Fig 11A, 7th bar; 11B, 4th bar); (*vi*) With the combined treatment, the Na⁺ balance remained similar to that with Ang-(3–4) alone in RBD rats (Fig 11C, 3rd *vs.* 4th bars). The overall comparison between Fig 11B and 11C allowed us to see how the nutritional status modifies the response of Na⁺ balance to Losartan and Ang-(3–4).

RBD rats had a lower [Na⁺]$_{pls}$ than the CTRL group, and it fell further with the chronic administration of Losartan, which also decreased [Na⁺]$_{pls}$ in the CTRL (Fig 12A). Ang-(3–4) alone decreased [Na⁺]$_{pls}$ only in CTRL rats (Fig 12B) and, in Los-treated rats, there was no effect of Ang-(3–4) administration irrespective of the nutritional status (Fig 12C). When one compares the effect of the combination Los+Ang-(3–4) with the untreated CTRL and RBD groups (Fig 12D), the decrease provoked by the drugs was 24% in the CTRL, whereas the response was lower (11%) in the RBD animals.

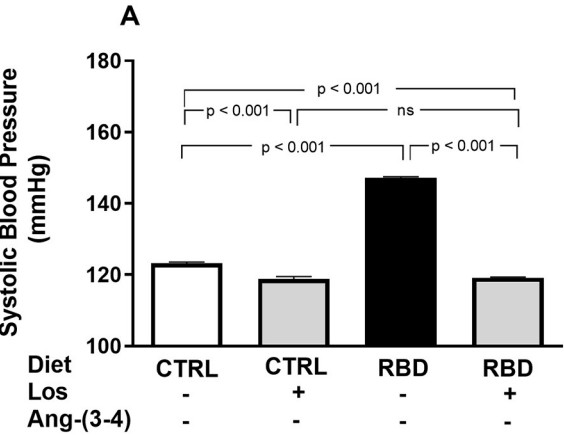

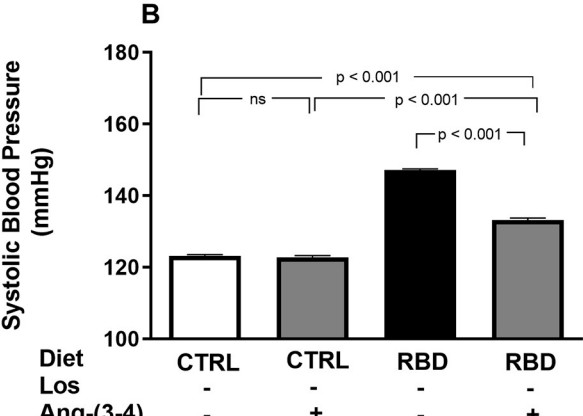

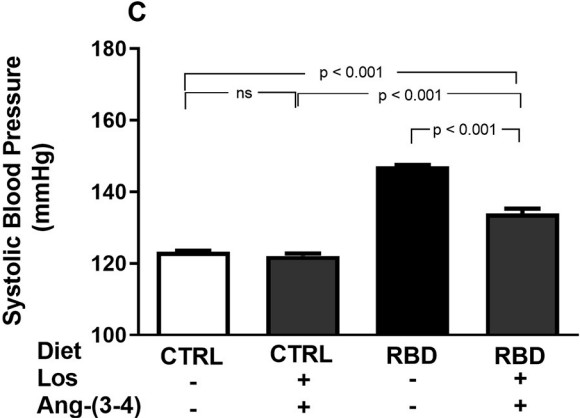

**Fig 7. Systolic blood pressure.** Losartan- and Ang-(3–4)-sensitive elevated systolic blood pressure (SBP) in undernourished rats. SBP was recorded in animals aged 91 or 92 days (in the case of Ang-(3–4)-treated rats) fed on CTRL or RBD diets. **(A)** Effects of Losartan. Comparison of SBP in CTRL and RBD rats without chronic administration of Losartan, and effects of Losartan administration (30 mg/kg body mass per day, from 28 to 92 days of life) to CTRL and RBD rats, as indicated on the *abscissa*. **(B)** Responses to Ang-(3–4). Effects of oral administration of a single dose of Ang-(3–4) (80 mg/kg body mass), at 91st day of life, to CTRL and RBD rats, as indicated on the *abscissa*. **(C)** Responses to Losartan+Ang-(3–4). Effects of combined administration of Losartan and Ang-(3–4) to CTRL and RBD rats, as shown on the *abscissa*. Bars are mean ± SEM; n = 10 (CTRL), n = 24 (RBD), n = 14 (CTRL

+Los), n = 14 (RBD+Los), n = 10 (CTRL+Ang-(3–4)), n = 14 (RBD+Ang-(3–4)), n = 10 (CTRL+Los+Ang-(3–4)), n = 9 (RBD+Los+Ang-(3–4)). Differences between means were analyzed using one-way ANOVA followed by Bonferroni's test for selected pairs. P values are given within the panels.

Since body Na+ balance is critically dependent on reabsorption of Na+ filtered by the renal glomeruli, a process which mostly (~75%) occurs in renal proximal tubules in an ATP-dependent manner [25, 26], we studied the influence of chronic undernutrition on the 2 Na+-transporting ATPases of proximal tubule cells: the ouabain-sensitive (Na++K+)ATPase and the ouabain-resistant Na+-ATPase [27, 28]. In RBD rats, (Na++K+)ATPase was 60% lower than in CTRL and insensitive to both Losartan and Ang-(3–4) (Fig 13A and 13B). In CTRL rats,

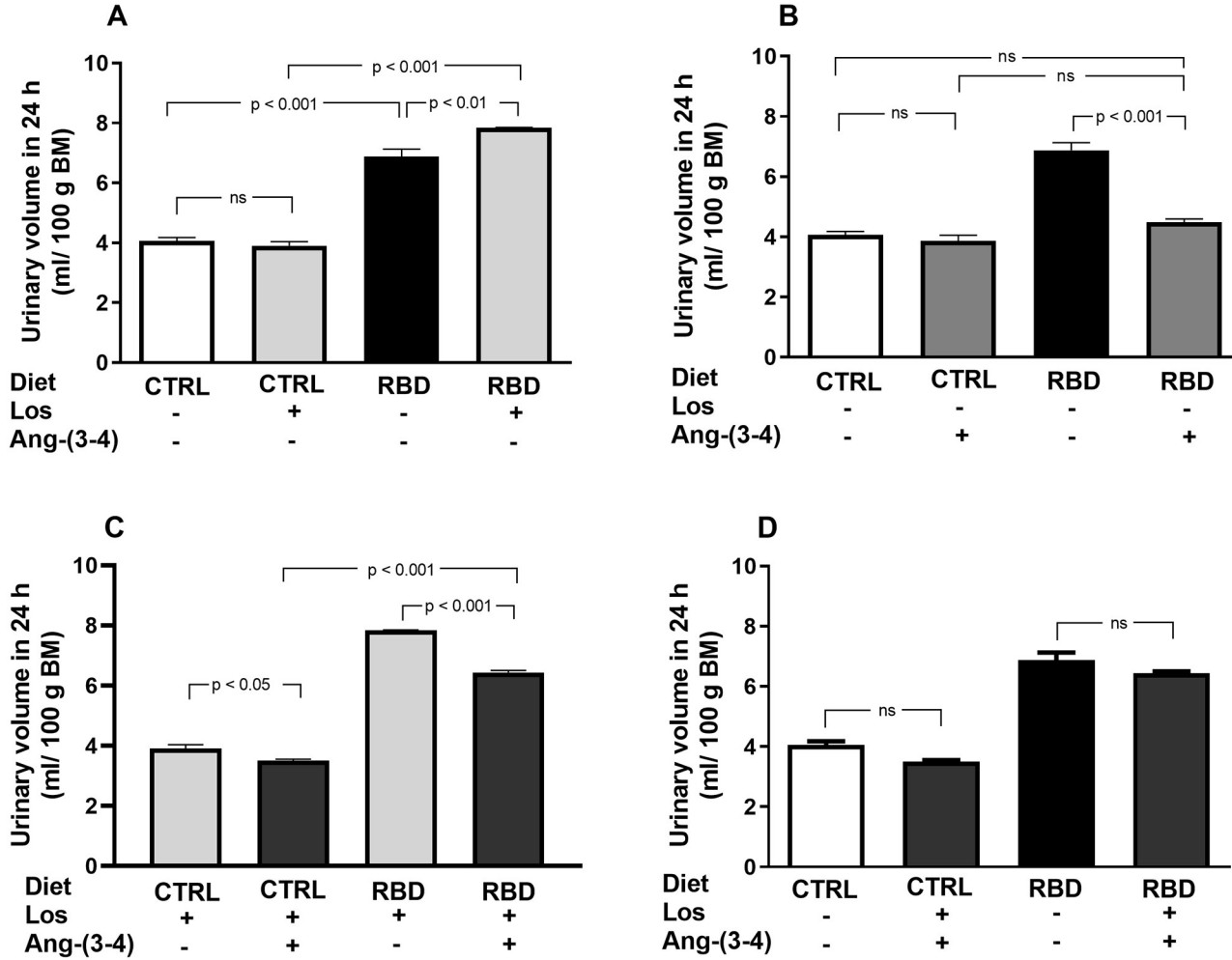

**Fig 8. Urinary volume in 24 h per 100 g BM.** Urine samples were collected between 90 and 91 days, except for Ang-(3–4)-treated rats, which were collected between 91 and 92 days. **(A)** Responses to Losartan. Comparison of the urinary volume of CTRL and RBD rats without chronic administration of Losartan, and effects of Losartan administration, as indicated on the *abscissa*. **(B)** Responses to Ang-(3–4). Effects of oral administration of a single dose of Ang-(3–4) on the urinary volume of CTRL and RBD rats, as indicated on the *abscissa*. **(C)** Responses to Ang-(3–4) in rats previously treated with Losartan. Effects of oral administration of a single dose of Ang-(3–4) on the urinary volume of CTRL and RBD rats previously treated with Losartan, as indicated on the *abscissa*. **(D)** Effects of combined Losartan+Ang-(3–4) administration. Comparison of urinary volume between untreated CTRL and RBD rats *vs.* CTRL and RBD rats that were chronically given Losartan and a single dose of Ang-(3–4), as indicated on the *abscissa*. Bars are mean ± SEM; n = 5 (CTRL), n = 8 (RBD), n = 5 (CTRL+Los), n = 6 (RBD+Los), n = 5 (CTRL+Ang-(3–4)), n = 8 (RBD+Ang-(3–4)), n = 5 (CTRL+Los+Ang-(3–4)), n = 6 (RBD+Los+Ang-(3–4)). Differences between means were analyzed using one-way ANOVA followed by Bonferroni's test for selected pairs. P values are indicated within the panels.

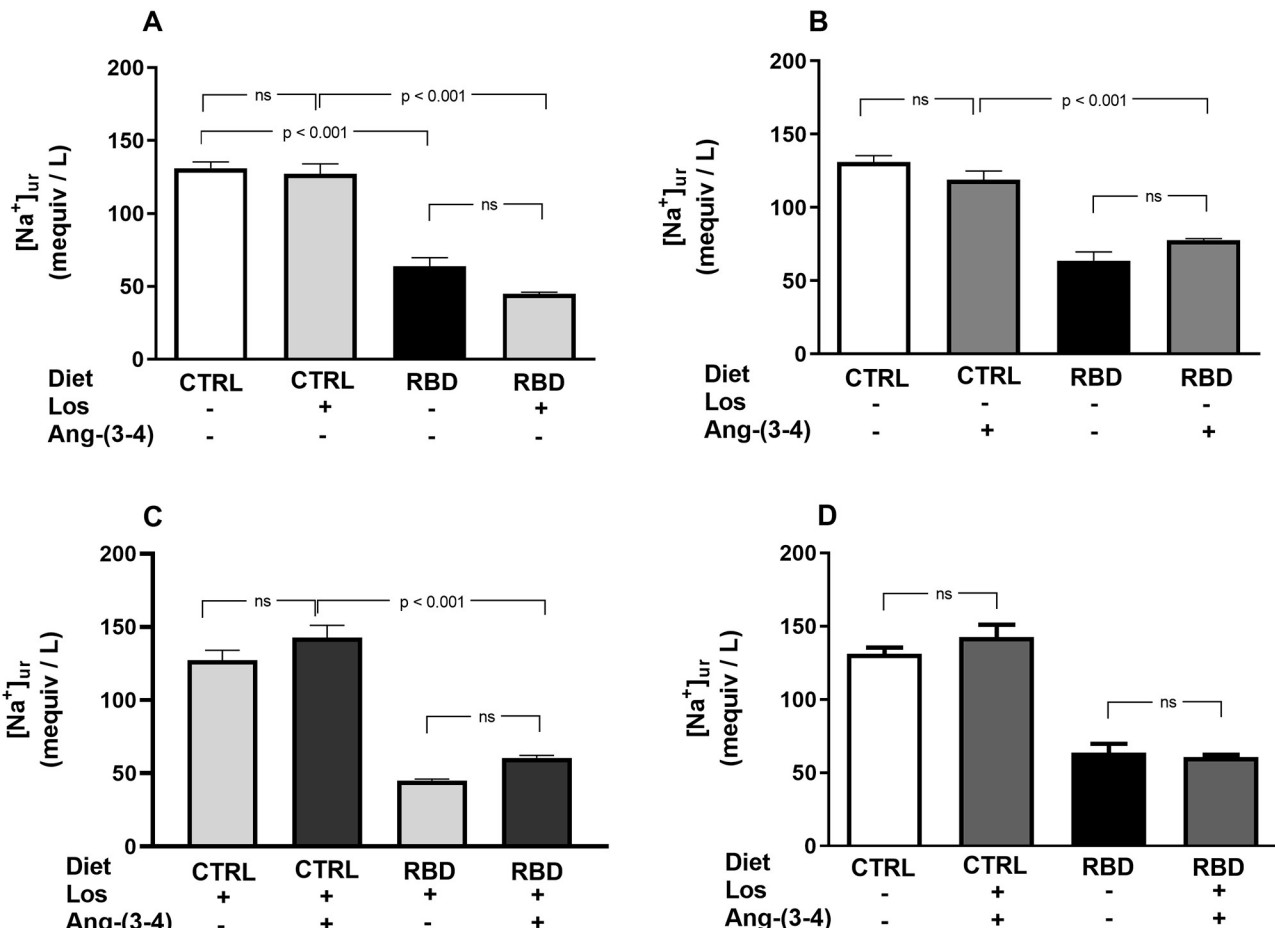

**Fig 9. Urinary Na$^+$ concentration ([Na$^+$]$_{ur}$).** Urine samples were collected between 90 and 91 days, except for Ang-(3–4)-treated rats, which were collected between 91 and 92 days. **(A)** Responses to Losartan. Comparison of [Na$^+$]$_{ur}$ in CTRL and RBD rats without chronic administration of Losartan, and effects of Losartan administration, as indicated on the *abscissa*. **(B)** Responses to Ang-(3–4). Effects of oral administration of a single dose of Ang-(3–4) on the [Na$^+$]$_{ur}$ of CTRL and RBD rats, as indicated on the *abscissa*. **(C)** Responses to Ang-(3–4) in rats previously treated with Losartan. Effects of oral administration of a single dose of Ang-(3–4) on the [Na$^+$]$_{ur}$ of CTRL and RBD rats previously treated with Losartan, as indicated on the *abscissa*. **(D)** Effects of combined Losartan+Ang-(3–4) administration. Comparison of [Na$^+$]$_{ur}$ between untreated CTRL and RBD rats *vs.* CTRL and RBD rats that were chronically given Losartan and a single dose of Ang-(3–4), as indicated on the *abscissa*. Bars are mean ± SEM; n = 5 (CTRL), n = 7 (RBD), n = 5 (CTRL+Los), n = 6 (RBD+Los), n = 5 (CTRL+Ang-(3–4)), n = 8 (RBD+Ang-(3–4)), n = 5 (CTRL+Los+Ang-(3–4)), n = 6 (RBD+Los +Ang-(3–4)). Differences between means were analyzed using one-way ANOVA followed by Bonferroni's test for selected pairs. P values are indicated within t he panels.

Losartan and Ang-(3–4), alone or in combination inhibited the (Na$^+$+K$^+$)ATPase by 25–30% (Fig 13A–13C), to a level that was similar to that found in the RBD group that had received the drugs in combination (compare the 2$^{nd}$ and 4$^{th}$ bars in Fig 13C). In contrast, RBD-induced undernutrition resulted in an accentuated upregulation of the ouabain-resistant Na$^+$-ATPase, which was inhibited by Losartan and Ang-(3–4), recovering the activity of the CTRL with the combined treatment (Fig 14A–14C). The Na$^+$-ATPase of the CTRL rats was completely insensitive to the drugs, alone or combined (Fig 14A–14C).

## Discussion

The central results of this study deal with the chronic ingestion of a multideficient diet (RBD) [5], which has led to the development of high blood pressure—the consequence of activation

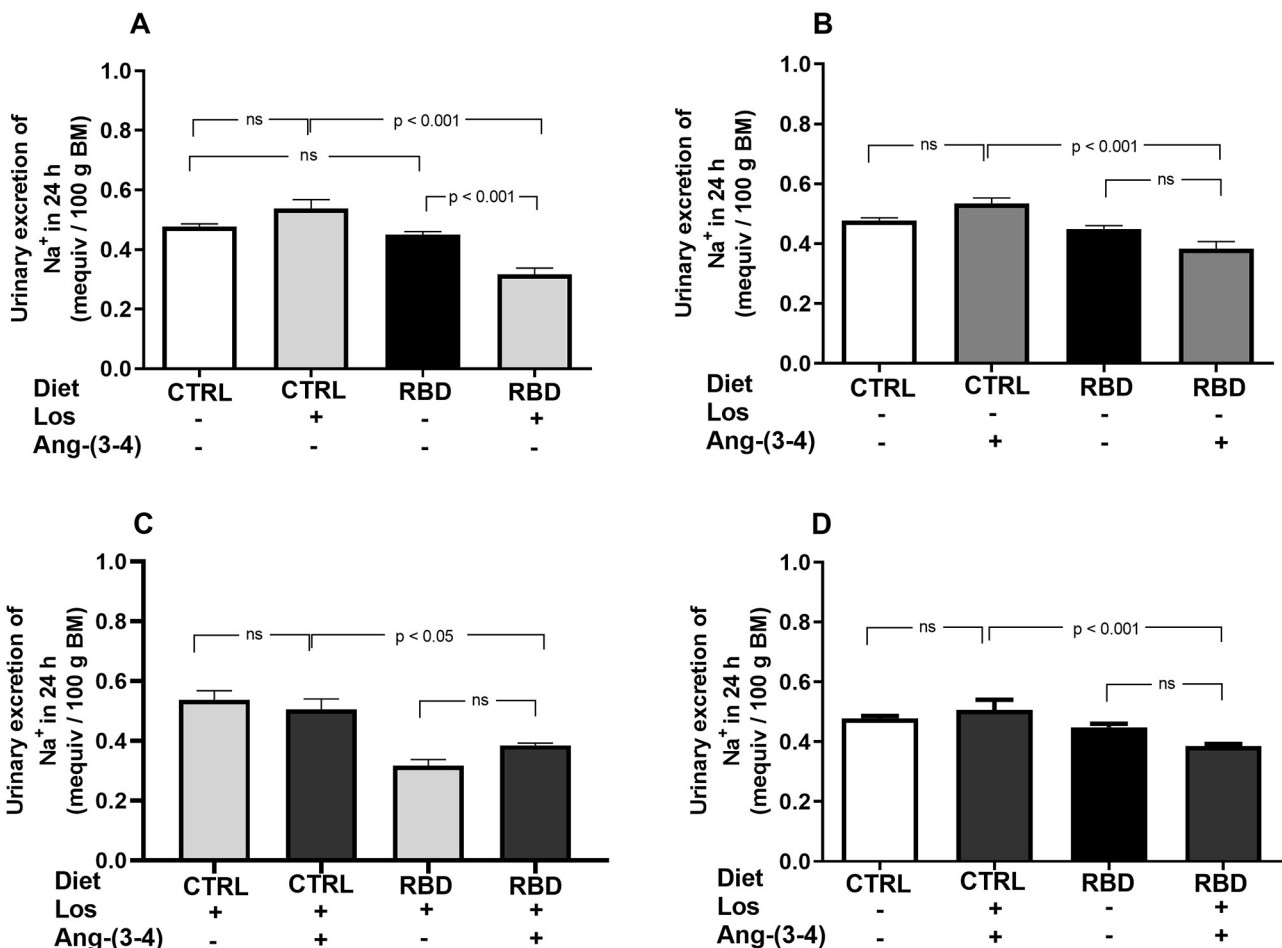

**Fig 10. Urinary Na+ excretion ($U_{Na}V$).** Urinary Na+ excretion in 24 h was calculated from $V_{ur}$ in 24 h/100 g BM and $[Na^+]_{ur}$ (Figs 8 and 9, respectively). (**A**) Responses to Losartan. Comparison of $U_{Na}V$ in CTRL and RBD rats without chronic administration of Losartan, and effects of Losartan administration, as indicated on the *abscissa*. (**B**) Responses to Ang-(3–4). Effects of oral administration of a single dose of Ang-(3–4) on the $U_{Na}V$ by CTRL and RBD rats, as indicated on the *abscissa*. (**C**) Responses to Ang-(3–4) in rats previously treated with Losartan. Effects of oral administration of a single dose of Ang-(3–4) on the $U_{Na}V$ by CTRL and RBD rats previously treated with Losartan, as indicated on the *abscissa*. (**D**) Effects of combined Losartan+Ang-(3–4) administration. Comparison of $U_{Na}V$ between untreated CTRL and RBD rats *vs*. CTRL and RBD rats that were chronically given Losartan and a single dose of Ang-(3–4), as indicated on the *abscissa*. Bars are mean ± SEM; n = 5 (CTRL), n = 7 (RBD), n = 5 (CTRL+Los), n = 6 (RBD+Los), n = 5 (CTRL+Ang-(3–4)), n = 8 (RBD+Ang-(3–4)), n = 5 (CTRL+Los+Ang-(3–4)), n = 6 (RBD+Los+Ang-(3–4)). Differences between means were analyzed using one-way ANOVA followed by Bonferroni's test for selected pairs. P values are given within the panels.

of the Losartan-sensitive Ang II⇒AT$_1$R axis of RAAS—along with an accentuated decrease in BM and changes in body Na+ handling. These alterations, which were totally or partially reversed by blockade of this axis with Losartan, or by activation of the counteracting Ang II⇒AT$_2$R axis by Ang-(3–4) [11, 12], occurred despite an increased caloric intake and a dietary Na+ content 20% lower than in the CTRL diet. Three points deserve consideration: (*i*) RBD administration is initiated immediately after weaning, a life period when occurs rapid and definitive developmental events [29, 30]; (*ii*) this diet mimics those consumed in vast regions of undeveloped countries [5, 31, 32]; and (*iii*) the extremely poor quality of the reduced protein content, the leading cause of morbidity and mortality in today's world [33, 34].

For years, the models investigating the mechanisms and processes affected by undernutrition were those based on diets with only low protein content [35, 36], which do not reflect the

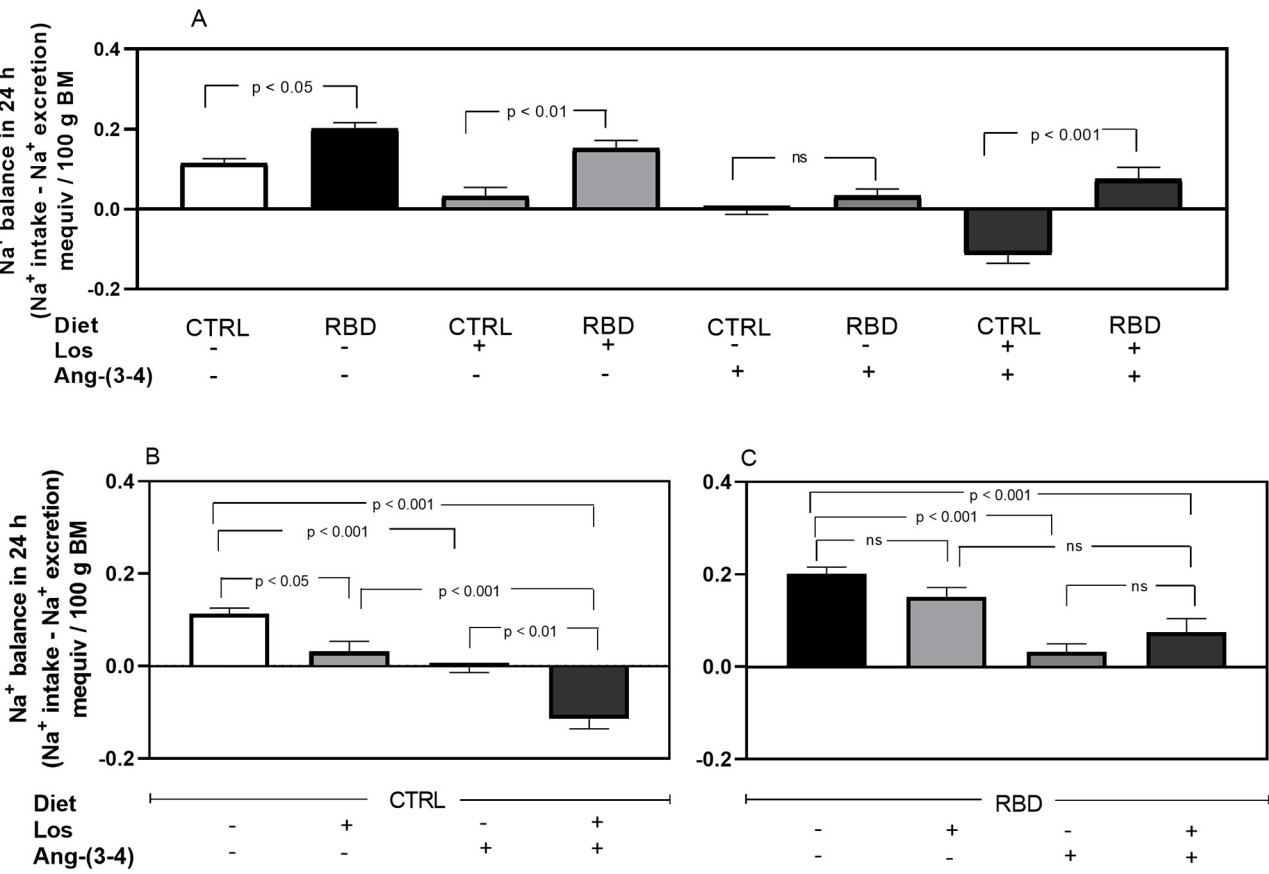

**Fig 11. Na+ balance.** Na+ balance (mequiv in 24 h/100 g BM) was calculated as the difference between Na+ intake and urinary Na+ excretion (Figs 5 and 10, respectively). The combinations of diets and treatments are indicated on the *abscissae*. **(A)** Effects of RBD administration without treatments or after treatment with Losartan, Ang-(3–4) or Losartan plus Ang-(3–4). **(B)** Effects of treatments in CTRL rats. **(C)** Effects of treatments in RBD rats. Bars are mean ± SEM; n = 5 (CTRL), n = 8 (RBD), n = 5 (CTRL+Los), n = 6 (RBD+Los), n = 5 (CTRL+Ang-(3–4)), n = 7 (RBD+Ang-(3–4)), n = 5 (CTRL+Los+Ang-(3–4)), n = 6 (RBD+Los+Ang-(3–4)). Differences between means were analyzed using one-way ANOVA followed by Bonferroni's test for selected pairs. P values are given within the panels.

situation of multi-deficiency found in the diets used in regions with endemic undernutrition [5, 34, 37]. Besides the quantitative deficiency in proteins, their sources in RBD—90% from beans and 10% from jerked meat—do not contain the quality required to preserve the normal pools of amino acids in the subcellular, cytosolic, and extracellular compartments [38]. The normal pools of amino acids depend on the total protein ingested in relation to lipids and carbohydrates, amino acid composition [39, 40], and adequate ingestion of vitamin B6 for proper synthesis of non-essential amino acids [41]. The multideficiency in RBD and the resulting amino acid disequilibrium [42] seem to provoke accentuated diminution in growth, despite a higher food and energy intake by the undernourished rats.

The associated mechanism for the onset of hypertension is the compensatory RBD-induced upregulation of RAAS, which also has a significant role in BM growth in early age under physiological conditions, as demonstrated by the growth profile curve when Losartan blocks the AT$_1$R (Fig 1). The development of hypertension at a juvenile age of 92 days (Fig 7), which is equivalent to 13–14 years in human lifespan [7], gives support to this hypothesis, which integrates both quantitative and qualitative dietary deficiencies with marked growth retardation in childhood, RAAS upregulation and the genesis of hypertension.

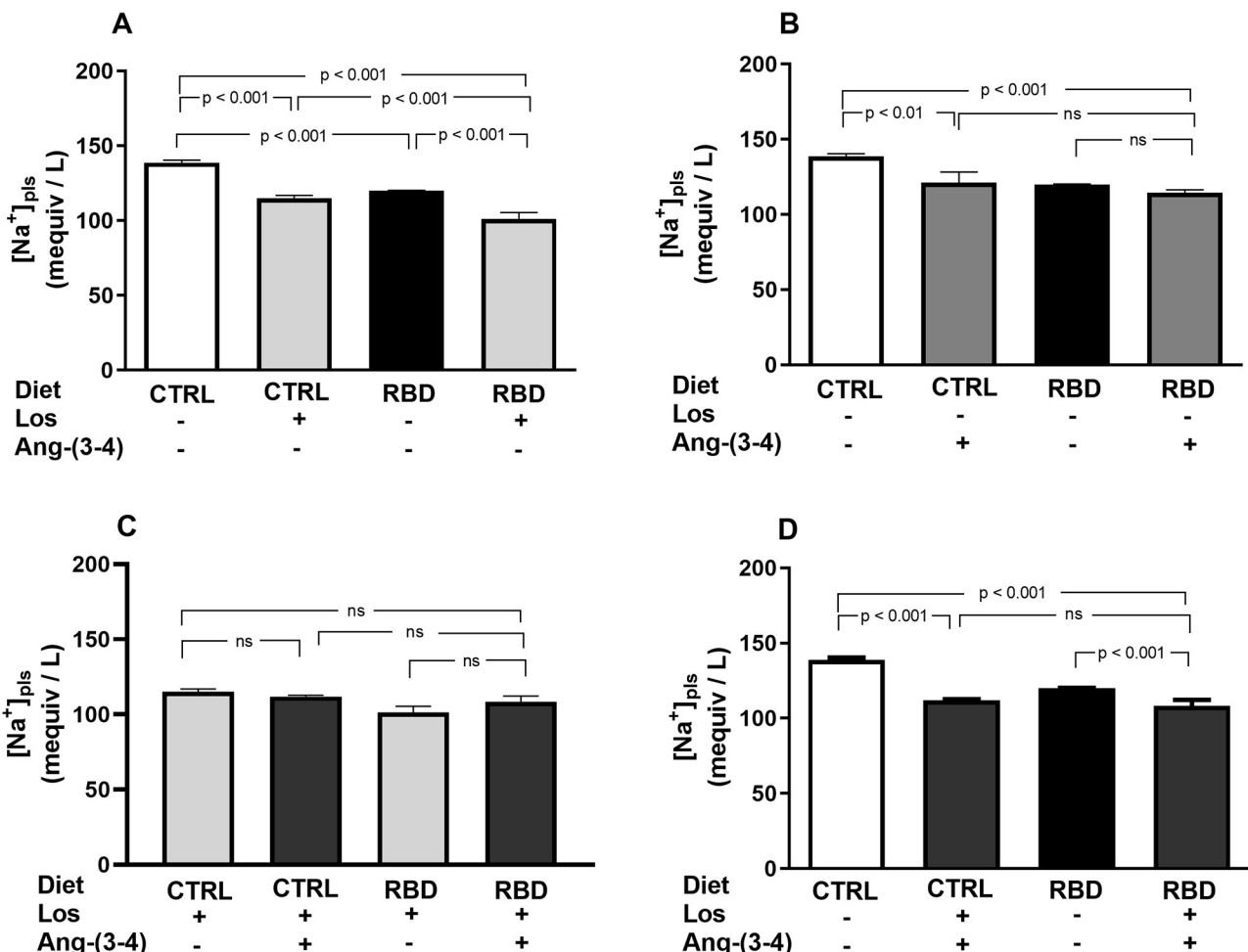

**Fig 12. Plasma Na⁺ concentration.** Plasma Na⁺ concentration ($[Na^+]_{pls}$) was determined in samples collected at 91 or 92 days (in the case of Ang-(3–4)-treated rats). **(A)** Responses to Losartan. Comparison of $[Na^+]_{pls}$ in CTRL and RBD rats without chronic administration of Losartan, and effects of Losartan administration, as indicated on the *abscissa*. **(B)** Responses to Ang-(3–4). Effects of oral administration of a single dose of Ang-(3–4) on the $[Na^+]_{pls}$ of CTRL and RBD rats, as indicated on the *abscissa*. **(C)** Responses to Ang-(3–4) in rats previously treated with Losartan. Effects of oral administration of a single dose of Ang-(3–4) on the $[Na^+]_{pls}$ of CTRL and RBD rats previously treated with Losartan, as indicated on the *abscissa*. **(D)** Effects of combined Losartan+Ang-(3–4) administration. Comparison of $[Na^+]_{pls}$ between untreated CTRL and RBD rats *vs*. CTRL and RBD rats that were chronically given Losartan and a single dose of Ang-(3–4), as indicated on the *abscissa*. Bars are mean ± SEM; n = 5 (CTRL), n = 15 (RBD), n = 5 (CTRL+Los), n = 5 (RBD+Los), n = 5 (CTRL+Ang-(3–4)), n = 8 (RBD+Ang-(3–4)), n = 5 (CTRL+Los+Ang-(3–4)), n = 5 (RBD+Los+Ang-(3–4)). Differences between means were analyzed using one-way ANOVA followed by Bonferroni's test for selected pairs. P values are given within the panels.

The importance of both RAAS axes also emerged from the influence of Losartan and Ang-(3–4) in different parameters. When food and energy intake are analyzed at a juvenile age, the roles of the Ang II⇒AT₁R and Ang II⇒AT₂R axes clearly emerge. Apart from the influence of Losartan in the BM evolution curves discussed above, which indicates the requirement of a functional Ang II⇒AT₁R axis for proper growth, several data show that the Ang II⇒AT₂R axis is also involved. Food intake was inhibited in both groups by a single administration of Ang-(3–4) (Fig 3B), which is a powerful antagonist of different Ang II effects in physiological and pathological conditions [9–12, 43], and energy intake was strongly inhibited by the combination of Losartan and Ang-(3–4) (Fig 4C), likely as the result of simultaneous blockade of the Ang II⇒AT₁R axis and stimulation of the Ang II⇒AT₂R axis. Perhaps, the effects of Ang-(3–4) rely on circuits of the central nervous system (CNS) that control hungry and, therefore,

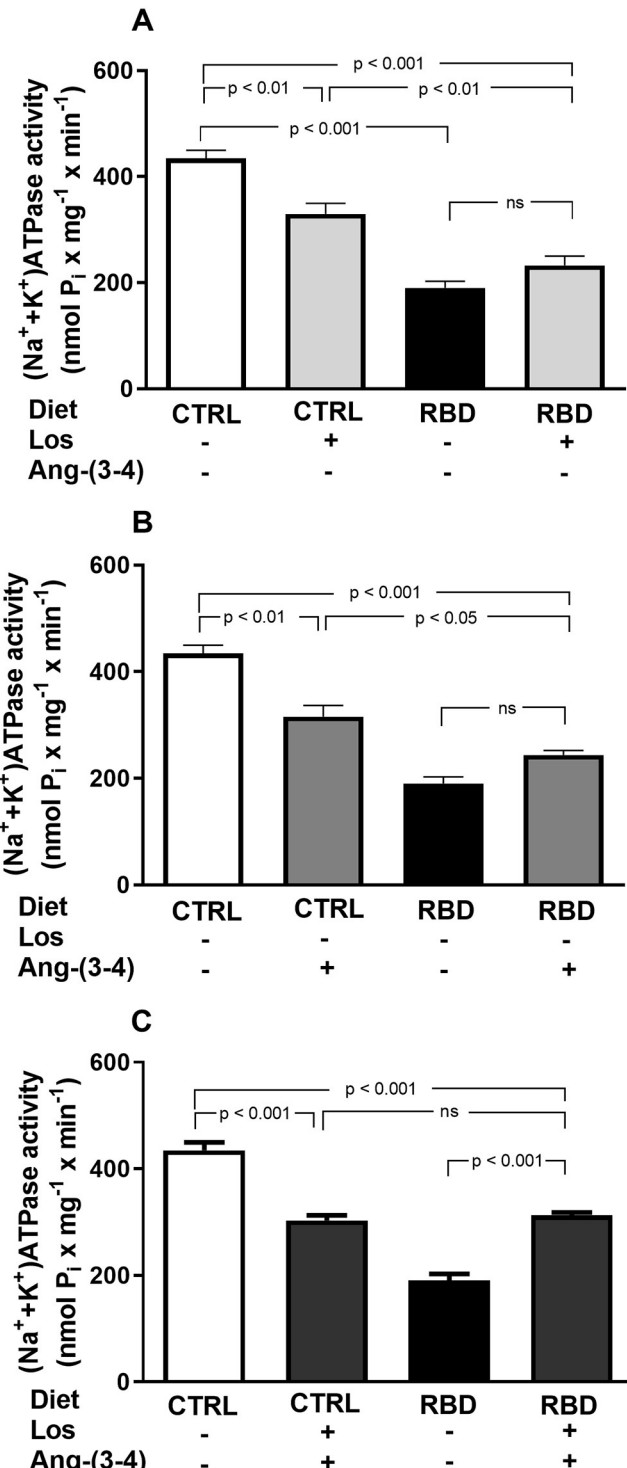

**Fig 13. Downregulation of the ouabain-sensitive (Na$^+$+K$^+$)ATPase from renal proximal tubule cells of chronically undernourished rats.** Determinations were carried out in plasma membrane-enriched preparations isolated from the outermost region of the renal cortex (*cortex corticis*) on day 92. **(A)** Effects of Losartan. Comparison of (Na$^+$+K$^+$) ATPase activity of CTRL and RBD rats without chronic administration of Losartan, and effects of Losartan administration (30 mg/kg body mass per day, from 28 to 92 days of life) to CTRL and RBD rats, as indicated on the *abscissa*. **(B)** Responses to Ang-(3–4). Effects of oral administration of a single dose of Ang-(3–4) (80 mg/kg body mass) on (Na$^+$+K$^+$)ATPase activity of CTRL and RBD rats, as indicated on the *abscissa*. **(C)** Responses to Losartan +Ang-(3–4). Effects of combined administration of Losartan and Ang-(3–4) on (Na$^+$+K$^+$)ATPase activity of CTRL and

RBD rats, as shown on the *abscissa*. Bars are mean ± SEM; n = 6 (CTRL), n = 6 (RBD), n = 6 (CTRL+Los), n = 6 (RBD+Los), n = 6 (CTRL+Ang-(3–4)), n = 6 (RBD+Ang-(3–4)), n = 6 (CTRL+Los+Ang-(3–4)), n = 5 (RBD+Los+Ang-(3–4)). Differences were assayed using one-way ANOVA followed by Bonferroni's test for selected pairs. P values are given within the panels.

food intake [44, 45]. The anorexigenic response to Ang-(3–4) in both groups is suggestive of an effect on the CNS of a peptide that can cross the blood-brain barrier, as recently demonstrated [46]. This central effect occurs in a way that seems to be dependent on the degree of upregulation of the Ang II⇒AT$_1$R axis because the effects of combined Losartan and Ang-(3–4) in food ingestion and energy intake disappeared in RBD rats (Figs 3C and 4C).

The Na$^+$ density data (Fig 6) show that the RBD rats ingest 59 kcal against 37 kcal per mequiv Na$^+$ of the CTRL or, in other words, that all undernourished groups incorporate more calories per Na$^+$ independent of the treatment. Since the RBD rats have the highest energy and Na$^+$ intake and have high blood pressure, we propose that Na$^+$ density rather than intake of salt alone is a determinant for the onset of hypertension in chronic undernutrition. These observations indicate that compensation for eating food containing small amounts of low-quality proteins becomes a key hypertensive mechanism [38]. In humans, the relationship between Na$^+$ and blood pressure varies when energy needs vary [24]. Additionally, analysis of the groups that received Ang-(3–4) alone or in combination with Losartan leads to the conclusion that simultaneous activation of the Ang II⇒AT$_2$R axis by Ang-(3–4) decreases Na$^+$ and energy intake. However, since the data from CTRL rats fell below the RBD data in a region of high Na$^+$ intake, and the normalization of blood pressure of the RBD rats is complete with Losartan but not with Ang-(3–4), we propose that the rapid response to Ang-(3–4) of the Ang II⇒AT$_2$R axis is impaired in chronic undernutrition.

The normotensive CTRL, CTRL+Los and RBD+Los rats deserve special consideration, which have their Na$^+$ density values in cluster ③ (Fig 6) and a comparable energy intake ranging 20–25 kcal in 24 h per 100 g BM, being a Na$^+$ intake much higher in CTRL and CTRL+Los rats than in RBD+Los rats. These observations show that blockade of the Ang II⇒AT$_1$R axis suffices for the normalization of SBP in undernourished animals, and also reinforce the proposal regarding the role of Ang II-regulated energy intake in the pathogenesis of undernutrition-associated hypertension.

Concerning possible interactions between the Ang II⇒AT$_1$R and Ang II⇒AT$_2$R axes in the pathogenesis of hypertension in RBD rats, the contrast between the total normalization of SBP with Losartan alone and the diminished effect of this drug after a single administration of Ang-(3–4) (Fig 7) also deserves special consideration. Three possibilities arise. First, since the effectiveness of chronic AT$_1$R blockade disappears when Ang-(3–4) activates the Ang II⇒AT$_2$R axis, it could be that the Ang-(3–4)-induced dissociation [10] of AT$_1$R/AT$_2$R dimers [47] results in Ang II⇒AT$_1$R-linked PKC-catalyzed abnormal phosphorylations of the contractile machinery [48] from the heart and aorta in a way that is resistant to Losartan. Second, a non-exclusive possibility is that increased Ang II⇒AT$_2$R-stimulated PKA upregulates PKC, a central component of Ang II⇒AT$_1$R signaling. This idea receives support regarding the tight connection between these two pathways [49, 50] in the heart of undernourished rats [38]. The third possibility would be that Ang-(3–4)-stimulated PKA activity switches G-coupling, leading, e.g., to G$_i$-dependent Losartan-insensitive activation to MAPK, as proposed 2 decades ago for several cardiovascular diseases [51]. MAPK is another central kinase in the heart and kidney of undernourished rats [38]. Since in CTRL rats, the SBP remained unmodified by Losartan, Ang-(3–4) or combined treatment, it seems clear that they only act in tissues with pro-

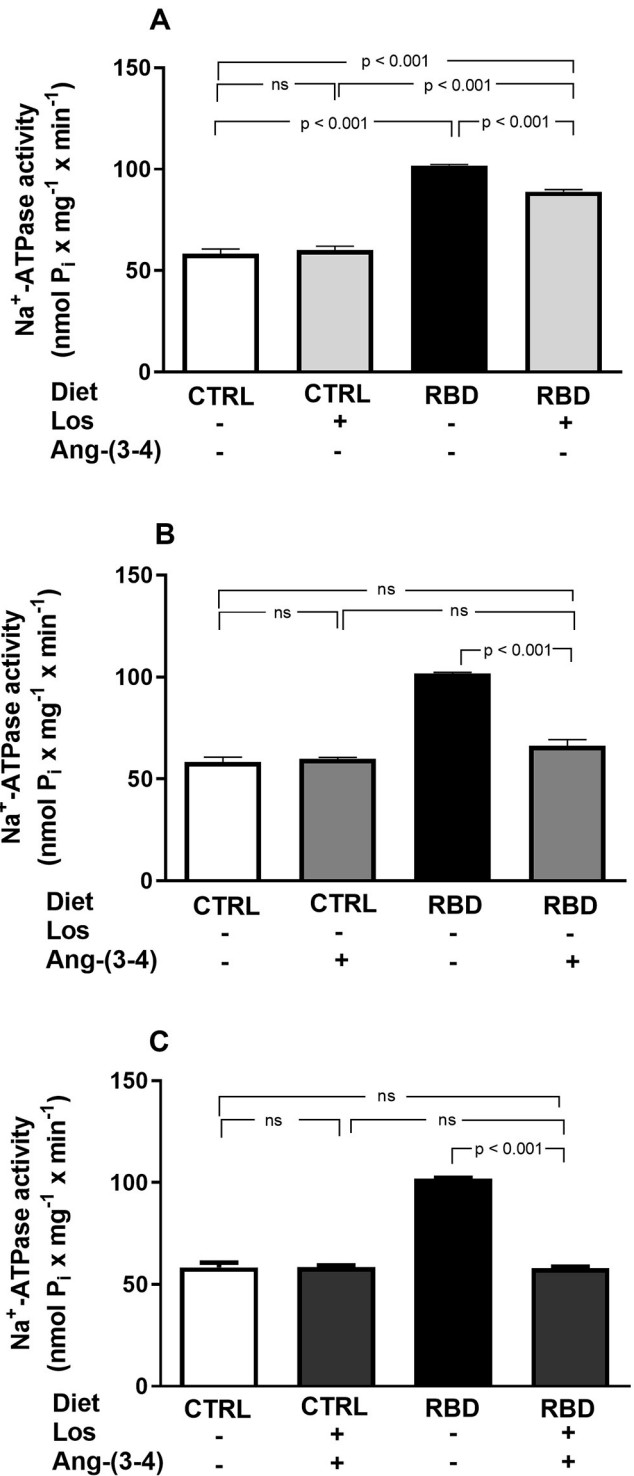

**Fig 14. Upregulation of the ouabain-resistant Na⁺-ATPase from renal proximal tubule cells of chronically undernourished rats.** Determinations were carried out in plasma membrane-enriched preparations isolated from the outermost region of the renal cortex (*cortex corticis*) on day 92. **(A)** Effects of Losartan. Comparison of Na⁺-ATPase activity of CTRL and RBD rats without chronic administration of Losartan, and effects of Losartan administration (30 mg/kg body mass per day, from 28 to 92 days of life) to CTRL and RBD rats, as indicated on the *abscissa*. **(B)** Responses to Ang-(3–4). Effects of oral administration of a single dose of Ang-(3–4) (80 mg/kg body mass) on Na⁺-ATPase activity of CTRL and RBD rats, as indicated on the *abscissa*. **(C)** Responses to Losartan+Ang-(3–4). Effects of combined administration of Losartan and Ang-(3–4) on Na⁺-ATPase activity of CTRL and RBD rats, as shown on the

*abscissa*. Bars are mean ± SEM; n = 4 in all conditions. Means were compared using one-way ANOVA followed by Bonferroni's test for selected pairs. P values are given within the panel.

hypertensive microenvironments [10, 13], i.e., in tissues with increased local activity of RAAS, as in the kidney of RBD rats [14].

The slightly increased Na⁺ intake (Fig 5) and the positive Na⁺ balance in RBD rats, which is ~100% higher than in CTRL rats (Fig 11), are indicative of Na⁺ accumulation that possibly occurred together with the onset of hypertension. Even though the RBD rats had expanded intravascular compartment [38], they are hyponatremic (Fig 12A). Thus, it may be that the positive $U_{Na}V$ of ~0.1 mequiv in 24 h/100 g BM above the level encountered in CTRL rats reflects Na⁺ accumulation occurring in an osmotically silent compartment, such as the dermis. Here, Na⁺ bound possibly colocalizes with the glycosaminoglycan scaffold as demonstrated in humans [52] and rodents [53]. The responses of Na⁺ balance to the treatments by Losartan and Ang-(3–4) in CTRL and RBD rats—especially the accentuated negative Na⁺ balance with the combined treatment—lead us to conclude that they rely on the inhibition of the Ang II⇒$AT_1R$ axis and the counteracting stimulation by Ang-(3–4) of the Ang II⇒$AT_2R$ axis [12]. This is another evidence that Ang-(3–4) acts as an antagonist of Ang II effects in a way that is modulated by the activity of local RAAS [10, 11]. Moreover, combined analysis of Na⁺ data reveals an intriguing feature: reduction in Na⁺ balance occurs as the result of decreased intake rather than from increased $U_{Na}V$, giving further support to the hypothesis that Ang-(3–4) modulates mechanisms at the level of CNS, after crossing the blood-brain barrier [46].

Besides the RAAS-associated systemic influence of undernutrition in Na⁺ balance leading to its progressive accumulation, one of the main tissue-based abnormal mechanisms of Na⁺ handling in RBD rats seems to rely on the functioning and regulation of renal Na⁺-transporting ATPases, where again RAAS has a central role [54, 55]. There are two Na⁺-transporting ATPases in renal proximal tubule cells: the Na⁺ pump sensitive to ouabain that is coupled to K⁺ transport in the opposite direction, and the second Na⁺ pump that is resistant to ouabain and not coupled to K⁺ fluxes [21, 28, 56–59]. RBD rats, the inhibition of proximal tubules (Na⁺+K⁺)ATPase, which is responsible for the bulk reabsorption of filtered Na⁺ [25], possibly represents the smaller amount of filtered salt load that needs to be recovered in the tubules of rats with very reduced body mass and, therefore, with reduced Na⁺-containing liquid compartments. In contrast, upregulated ouabain-resistant Na⁺-ATPase responsible for the fine-tuning of proximal Na⁺ reabsorption [27, 28] is likely to be an essential mechanism involved in the increased positive Na⁺ balance depicted in RBD rats. Since there is no increase in $U_{Na}V$ by treatments with Losartan and Ang-(3–4), normalization of the proximal tubule Na⁺-ATPase by the 2 compounds may be ascribed to the restoration of the fine-tune reabsorption at the level of proximal tubules rather than a contribution for overall recovery of the normal and bulk Na⁺ balance.

## Conclusion

This study provides evidence that the chronic administration of a multideficient diet with a low content of protein of very poor quality is the primary cause—rather than excess Na⁺–in the pathogenesis of hypertension in undernourished rats, by simultaneously targeting the Ang II⇒$AT_1R$ axis of local RAAS in the kidney, the central nervous system (especially centers of food satiety, Na⁺ hungry control and cardiovascular regulation), and possibly bodily Na⁺ distribution and structural modifications of the cardiovascular system itself. Furthermore, these results give support to the view that the antagonism of the Ang II⇒$AT_1R$ axis by the Ang

II$\Rightarrow$AT$_2$R axis within the RAAS is mediated, at least in part, by central and peripheral actions of Ang-(3–4), the potent allosteric enhancer of AT$_2$R that increases the affinity of AT$_2$R for Ang II [12].

## Acknowledgments

The authors express special thanks to the Laboratory of Food Analysis and Processing, Josué de Castro Institute of Nutrition at Federal University of Rio de Janeiro. The finally submitted version of the manuscript was prepared by BioMedES Ltd., UK (www.biomedes.biz).

## Author Contributions

**Conceptualization:** Amaury Pereira-Acácio, Humberto Muzi-Filho, Adalberto Vieyra.

**Data curation:** Amaury Pereira-Acácio, João P. M. Veloso-Santos, Gloria Costa-Sarmento.

**Formal analysis:** Amaury Pereira-Acácio, João P. M. Veloso-Santos, Luiz F. Nossar, Gloria Costa-Sarmento, Humberto Muzi-Filho, Adalberto Vieyra.

**Funding acquisition:** Amaury Pereira-Acácio, Humberto Muzi-Filho, Adalberto Vieyra.

**Investigation:** Amaury Pereira-Acácio, João P. M. Veloso-Santos, Luiz F. Nossar, Gloria Costa-Sarmento.

**Methodology:** Amaury Pereira-Acácio, João P. M. Veloso-Santos, Luiz F. Nossar, Gloria Costa-Sarmento, Humberto Muzi-Filho.

**Resources:** Adalberto Vieyra.

**Supervision:** Adalberto Vieyra.

**Validation:** Amaury Pereira-Acácio, João P. M. Veloso-Santos, Gloria Costa-Sarmento, Humberto Muzi-Filho, Adalberto Vieyra.

**Visualization:** Amaury Pereira-Acácio, Humberto Muzi-Filho, Adalberto Vieyra.

**Writing – original draft:** Amaury Pereira-Acácio, João P. M. Veloso-Santos, Humberto Muzi-Filho, Adalberto Vieyra.

**Writing – review & editing:** Amaury Pereira-Acácio, João P. M. Veloso-Santos, Humberto Muzi-Filho, Adalberto Vieyra.

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
