## [Decision Letter · Decision Letter 0]

30 May 2022

PONE-D-22-09686Angiotensin-(3–4) normalizes the elevated arterial blood pressure and abnormal Na+/energy handling associated with chronic undernutrition by counteracting the effects mediated by type 1 angiotensin II receptorsPLOS ONE

Dear Dr. Vieyra

Thank you for submitting your manuscript to PLOS ONE. After careful consideration, we feel that it has merit but does not fully meet PLOS ONE’s publication criteria as it currently stands. Therefore, we invite you to submit a revised version of the manuscript that addresses the points raised during the review process.

We look forward to receiving your revised manuscript.

Kind regards,

Xianwu Cheng, M.D., Ph.D., FAHA

Academic Editor

PLOS ONE

Journal Requirements:

Additional Editor Comments:

The both reviewers have concerned statistical analysis. Thus, the authors should address all of these comments satisfactory before resubmission.

Reviewers' comments:

Reviewer's Responses to Questions

**Comments to the Author**

1. Is the manuscript technically sound, and do the data support the conclusions?

Reviewer #1: Partly

Reviewer #2: Yes

2. Has the statistical analysis been performed appropriately and rigorously? 

Reviewer #1: No

Reviewer #2: Yes

3. Have the authors made all data underlying the findings in their manuscript fully available?

Reviewer #1: No

Reviewer #2: Yes

4. Is the manuscript presented in an intelligible fashion and written in standard English?

Reviewer #1: Yes

Reviewer #2: Yes

5. Review Comments to the Author

Reviewer #1: General Comments

In this study, the authors investigated the mechanisms of bodily Na+ handling and high systolic blood pressure in the rat model feeding a multideficient diet. They used two renin-angiotensin-aldosterone system (RAAS)-related drugs, losartan as an antagonist of AT1R, and Ang-(3–4) as an “allosteric enhancer” of AT2R .

They concluded that the chronic administration of a multideficient diet with a low content of protein and very poor quality is the primary cause – rather than excess Na+ – in the pathogenesis of hypertension in this rat model, by simultaneously targeting the Ang II�AT1R axis of local RAAS in kidney, the central nervous system (especially centers of food satiety, Na+ hungry control and cardiovascular regulation), and possibly bodily Na+ distribution and structural modifications of the cardiovascular system itself.

Specific comments.

1) P values are presented as four decimal places, which is meaningless. This is animal study, therefore P<0.01, P<0.001 or two decimal places is enough. See the article in PloS One reported from your laboratory (PLoS One. 2014 Jul 1;9(7):e100410).

2) Presentation of Figures are not appropriate. In Fig 1, for example, the same values of CTR (-/-)and RBD (-/-) columns show 4 columns in the same figure.

3) Also, the authors misunderstand ANOVA. ANOVA is applied to estimate the strictly correct variance of the parameter of interest to avoid screwed variances. Therefore, all groups in Fig1 must be put in the same ANOVA for calculation. The authors cannot use Student t-test for analysis to compare two groups of several groups.

4) Since there are no restrictions on the number of figures, it is better not to use supplementary figures.

5) Did the authors check the yield or purity of plasma membrane-rich fraction prepared for each ATPase activity from kidney proximal tubule cells among groups.

6) Manuscript text should be double-spaced according to instruction for authors.

Reviewer #2: In the current study, the authors investigated the mechanism of hypertension associated with chronic undernutrition. This study provides evidence that the chronic administration of a multi-deficient diet with a low content of protein of very poor quality is the primary cause – rather than excess Na+ in the pathogenesis of hypertension in undernourished rats, by simultaneously targeting the Ang II-AT1R axis of local RAAS in kidney, the central nervous system, and possibly bodily Na+ distribution and structural modifications of the cardiovascular system itself. I believe this study sheds light on the role of Ang II-AT1R axis and Ang II-AT2R axis in hypertension under the chronic undernutrition. However, I have some questions that needed to be answered.

1. Do the changes caused by RBD in the Wistar rats resemble that in human being, including Body weight, energy intake, Na+ ingestion，Na+ excretion, hypertension extent ?

2. Figure 7，the elevated SBP is almost normalized by Losartan in RBD group. Ang-(3-4）also has an hypotensive effect in RBD. Why the SBP seemed to increase SBP in the combination therapy group (Lorsartan+Ang-(3-4）) as compared to the single therapy group (Lorsartan or Ang-(3-4）) under RBD？

3. In figure 11-12，the ouabain-sensitive Na+ ATPase and the ouabain-resistant Na+ ATPase activities changes differently under RBD. What’s the different roles between the two kinds of Na+ ATPase? Please give more explanation in the Discussion.

6. PLOS authors have the option to publish the peer review history of their article (what does this mean?). If published, this will include your full peer review and any attached files.

Reviewer #1: No

Reviewer #2: No

---

## [Author Response · Author response to Decision Letter 0]

29 Jun 2022

Please, see uploaded file "Response to Reviewers".

---

## [Decision Letter · Decision Letter 1]

21 Jul 2022

PONE-D-22-09686R1Angiotensin-(3–4) normalizes the elevated arterial blood pressure and abnormal Na+/energy handling associated with chronic undernutrition by counteracting the effects mediated by type 1 angiotensin II receptorsPLOS ONE

Dear Dr. Vieyra 

Thank you for submitting your manuscript to PLOS ONE. After careful consideration, we feel that it has merit but does not fully meet PLOS ONE’s publication criteria as it currently stands. Therefore, we invite you to submit a revised version of the manuscript that addresses the points raised during the review process.

We look forward to receiving your revised manuscript.

Kind regards,

Xianwu Cheng, M.D., Ph.D., FAHA

Academic Editor

PLOS ONE

Journal Requirements:

Additional Editor Comments:

None.

Reviewers' comments:

Reviewer's Responses to Questions

**Comments to the Author**

1. If the authors have adequately addressed your comments raised in a previous round of review and you feel that this manuscript is now acceptable for publication, you may indicate that here to bypass the “Comments to the Author” section, enter your conflict of interest statement in the “Confidential to Editor” section, and submit your "Accept" recommendation.

Reviewer #1: All comments have been addressed

Reviewer #2: All comments have been addressed

2. Is the manuscript technically sound, and do the data support the conclusions?

Reviewer #1: Yes

Reviewer #2: Yes

3. Has the statistical analysis been performed appropriately and rigorously? 

Reviewer #1: I Don't Know

Reviewer #2: Yes

4. Have the authors made all data underlying the findings in their manuscript fully available?

Reviewer #1: Yes

Reviewer #2: Yes

5. Is the manuscript presented in an intelligible fashion and written in standard English?

Reviewer #1: Yes

Reviewer #2: Yes

6. Review Comments to the Author

Reviewer #1: The revised version has been significantly improved.

RE: P values in the Figures.

"P> 0.05" is not appropriate because the authors declare that "significant difference was set to P <0.05". The authors should use P ≥ 0.05 instead of P > 0.05. If they simply want to express that there is no statistical difference, the reviewer recommends using NS (not significant). Otherwise, each figure will be confused. In other articles, the P value is expressed to the second decimal place even if there is no significant difference.

Reviewer #2: The authors have adequately addressed my comments raised in a previous round of review. I believe that this manuscript is now acceptable for publication.

7. PLOS authors have the option to publish the peer review history of their article (what does this mean?). If published, this will include your full peer review and any attached files.

Reviewer #1: No

Reviewer #2: No

---

## [Author Response · Author response to Decision Letter 1]

22 Jul 2022

Please, see the uploaded file "Response to Reviewers R1".

---

## [Decision Letter · Decision Letter 2]

8 Aug 2022

Angiotensin-(3–4) normalizes the elevated arterial blood pressure and abnormal Na+/energy handling associated with chronic undernutrition by counteracting the effects mediated by type 1 angiotensin II receptors

PONE-D-22-09686R2

Dear Dr. Vieyra

We’re pleased to inform you that your manuscript has been judged scientifically suitable for publication and will be formally accepted for publication once it meets all outstanding technical requirements.

Kind regards,

Xianwu Cheng, M.D., Ph.D., FAHA

Academic Editor

PLOS ONE

Additional Editor Comments (optional):

All concerns have been addressed by the authors.

Reviewers' comments:

Reviewer's Responses to Questions

**Comments to the Author**

1. If the authors have adequately addressed your comments raised in a previous round of review and you feel that this manuscript is now acceptable for publication, you may indicate that here to bypass the “Comments to the Author” section, enter your conflict of interest statement in the “Confidential to Editor” section, and submit your "Accept" recommendation.

Reviewer #1: All comments have been addressed

Reviewer #2: All comments have been addressed

2. Is the manuscript technically sound, and do the data support the conclusions?

Reviewer #1: Yes

Reviewer #2: Yes

3. Has the statistical analysis been performed appropriately and rigorously? 

Reviewer #1: I Don't Know

Reviewer #2: Yes

4. Have the authors made all data underlying the findings in their manuscript fully available?

Reviewer #1: Yes

Reviewer #2: Yes

5. Is the manuscript presented in an intelligible fashion and written in standard English?

Reviewer #1: Yes

Reviewer #2: Yes

6. Review Comments to the Author

Reviewer #1: The revised version is corrected as the reviewer point outed.　The figures are much easier understand.

Reviewer #2: All the concerns have been fully addressed. I think the current version can be accepttable for publication now.

7. PLOS authors have the option to publish the peer review history of their article (what does this mean?). If published, this will include your full peer review and any attached files.

Reviewer #1: No

Reviewer #2: No

---

## [Editor Report · Acceptance letter]

11 Aug 2022

PONE-D-22-09686R2 

Angiotensin-(3–4) normalizes the elevated arterial blood pressure and abnormal Na^+^/energy handling associated with chronic undernutrition by counteracting the effects mediated by type 1 angiotensin II receptors 

Dear Dr. Vieyra:

I'm pleased to inform you that your manuscript has been deemed suitable for publication in PLOS ONE. Congratulations! Your manuscript is now with our production department. 

Kind regards, 

on behalf of

Associate Prof. Xianwu Cheng 

Academic Editor

PLOS ONE